# Optimal evolutionary decision-making to store immune memory

Oskar H Schnaack[1,2], Armita Nourmohammad[1,2,3]*

[1]Max Planck Institute for Dynamics and Self-organization, Göttingen, Germany; [2]Department of Physics, University of Washington, Seattle, United States; [3]Fred Hutchinson Cancer Research Center, Seattle, United States

**Abstract** The adaptive immune system provides a diverse set of molecules that can mount specific responses against a multitude of pathogens. Memory is a key feature of adaptive immunity, which allows organisms to respond more readily upon re-infections. However, differentiation of memory cells is still one of the least understood cell fate decisions. Here, we introduce a mathematical framework to characterize optimal strategies to store memory to maximize the utility of immune response over an organism's lifetime. We show that memory production should be actively regulated to balance between affinity and cross-reactivity of immune receptors for an effective protection against evolving pathogens. Moreover, we predict that specificity of memory should depend on the organism's lifespan, and shorter lived organisms with fewer pathogenic encounters should store more cross-reactive memory. Our framework provides a baseline to gauge the efficacy of immune memory in light of an organism's coevolutionary history with pathogens.

## Introduction

Adaptive immunity in vertebrates develops during the lifetime of an organism to battle a multitude of evolving pathogens. The central actors in our adaptive immune system are diverse B- and T-cells, whose unique surface receptors are generated through genomic rearrangement, mutation, and selection (*Janeway et al., 2005*). The diversity of receptors allows the immune system to mount specific responses against diverse pathogens. B-cell receptors (BCRs) in particular can specialize through a process of affinity maturation, which is a form of *somatic Darwinian evolution* within an individual to enhance the affinity of BCRs to pathogens. Several rounds of somatic mutation and selection during affinity maturation can increase binding affinities of BCRs up to 10,000 fold (*Victora and Nussenzweig, 2012*; *Meyer-Hermann et al., 2012*).

Beside receptor diversity, immune cells also differentiate and specialize to take on different roles, including plasma B-cells, which are antibody factories, effector T-cells, which can actively battle infections, or memory cells. Memory responses are highly efficient because memory cells can be reactivated faster than naive cells and can mount a more robust response to an infection (*McHeyzer-Williams et al., 2000*; *Tangye et al., 2003*; *Tangye and Hodgkin, 2004*; *Moens et al., 2016*). Memory generation is a form of cell fate decision in the immune system, which can occur at different stages of an immune response. In B-cells, activated naive cells can differentiate into antibody-secreting long-lived plasma cells, a T-cell-independent un-hypermutated memory cells, or they can initiate a germinal center (*Goodnow et al., 2010*). B-cells that enter germinal centers differentiate during affinity maturation into high-affinity plasma cells or T-cell-dependent long-lived memory cells that circulate in the blood for antigen surveillance; see schematic *Figure 1*.

The basis for differentiation of B-cells into memory, especially during affinity maturation, is among the least understood in cell fate decision-making in the immune system (*Goodnow et al., 2010*). A long-standing view was that memory is continuously produced during affinity maturation

*For correspondence:
armita@uw.edu

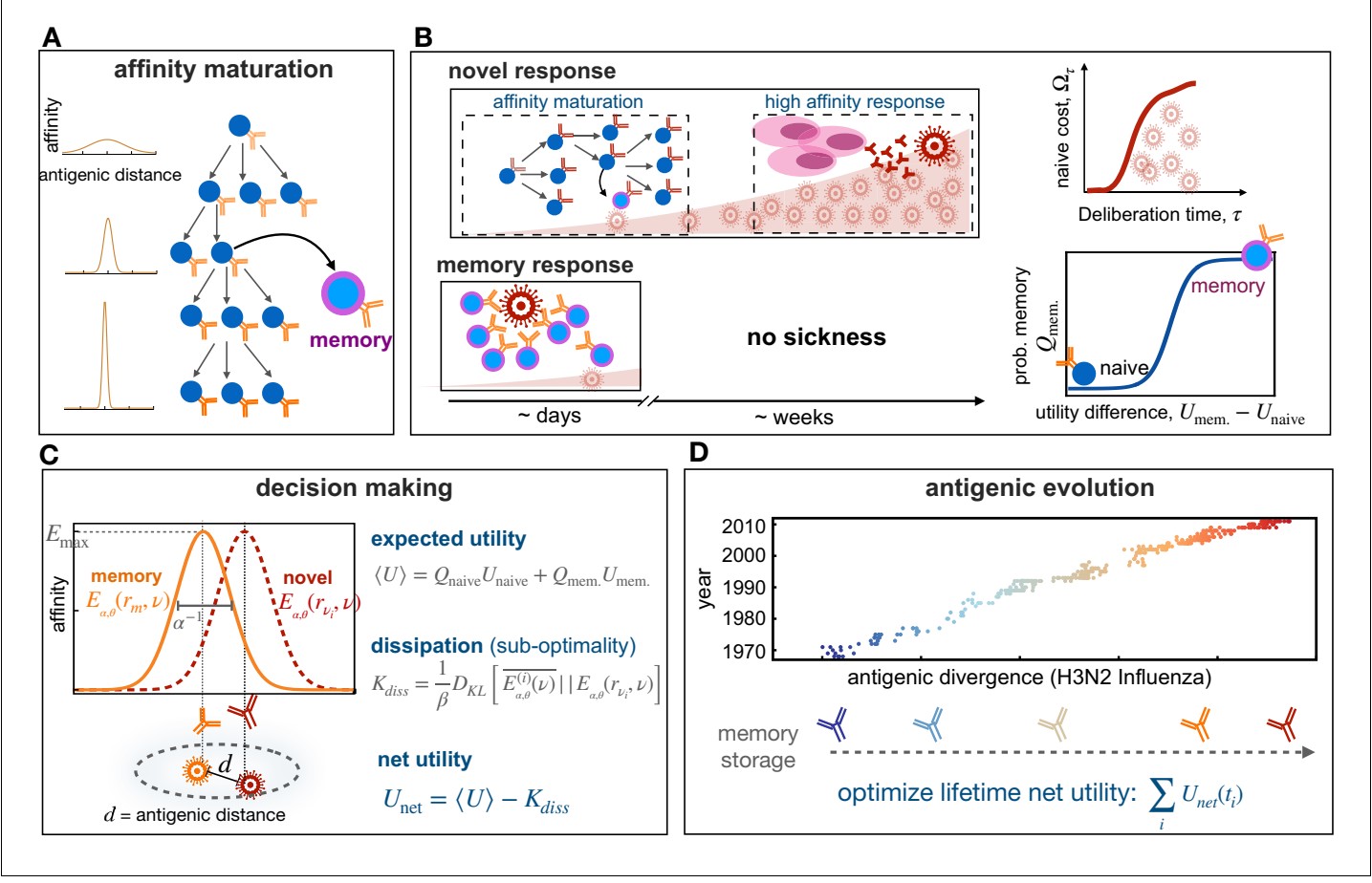

**Figure 1.** Immune memory or naive response upon infection. (A) Schematic shows affinity maturation in germinal centers(right), where B-cell receptors acquire mutations and undergo selection, resulting in an increase in their affinity to an antigen (from light to dark receptors), indicated by the sharpening of receptors' affinity profiles (on left). (B) Upon infection, the immune system can initiate a novel response (top) or a memory response (bottom). A novel B-cell response could involve affinity maturation to generate memory or high-affinity plasma cells (pink) that can secrete antibodies to battle the pathogen. A novel response can take 1–2 weeks, during which pathogen can replicate within a host and a patient can show symptoms from the disease (top, left). During this time, the proliferation of pathogens within a host incurs a cost associated with a naive response $\Omega_\tau$, which is a monotonic function of the deliberation time $\tau$ (top, right). If the host carries memory from a previous infection or vaccination (bottom), the immune system can robustly and rapidly activate a memory response to battle the infection. The probability to mount such memory response $Q_{\text{mem.}}$ depends non-linearly on the relative utilities of memory versus naïve responses against a given infection $\Delta U = U_{\text{mem.}} - U_{\text{naive}}$ (bottom, right). (C) Affinity profile $E_{\alpha,\theta}(r_m, v) \sim \alpha \exp[-(\alpha d)^\theta]$ of a memory receptor $r_m$ is shown in orange as a function of the distance $d = \|v_r^* - v\|$ in the antigenic shape space, between the receptor's cognate antigen $v_r^*$ (orange) and an evolved novel target $v_i$ (red). The affinity of a receptor decays with increasing distance between targets and its cognate antigen. The antigenic range over which a receptor is reactive inversely depends on its specificity $\alpha$. The shape of the binding profile is tuned by the factor $\theta$, here shown for $\theta = 2$. The expected binding profile $\overline{E_{\alpha,\theta}^{(i)}(v)}$ and the expected utility $\langle U \rangle$ for an immune response are weighted averages of these quantities over memory and naïve responses. The Kullback-Leibler distance between the expected profile $\overline{E_{\alpha,\theta}^{(i)}(v)}$ and the profile centered around the infecting antigen $E_{\alpha,\theta}(r_{v_i}, v)$, in units of the deliberation factor $\beta$, defines the sub-optimality of a response, that is,, dissipation $K_{diss}$ (**Equation 1**). The net utility $U_{\text{net}}$ measures the goodness of a decision to mount a memory vs. naive response against an infection (**Equation 2**). (D) Antigenic evolution of the H3N2 influenza virus is shown over 40 years along its first (most variable) antigenic dimension (data from **Bedford et al., 2014**). The decision of an immune system to utilize memory or to mount a novel response (B,C) is determined by the specificity $\alpha$ of receptors and the deliberation factor $\beta$. We characterize the optimal immune strategies $(\alpha^*, \beta^*)$ by maximizing the total net utility of immune responses against pathogens with different antigenic divergences, experienced over the lifetime of an organisms (**Equation 3**).

(**Blink et al., 2005**). Memory receptors often have lower affinity compared to plasma cells (**Smith et al., 1997**), and therefore, if memory B-cells were to be generated continuously it should be able to proliferate without strong affinity-dependent selection (**Goodnow et al., 2010**; **Victora and Nussenzweig, 2012**). However, recent experiments indicate that memory

differentiation is highly regulated (*Paus et al., 2006*; *Weisel et al., 2016*; *Shinnakasu et al., 2016*; *Recaldin and Fear, 2016*; *Shinnakasu and Kurosaki, 2017*; *Viant et al., 2020*), reflecting a temporal switch in germinal centers that preferentially produces memory at early stages and plasma at later stages of affinity maturation (*Weisel et al., 2016*). This active regulation introduces an affinity-dependent cell fate decision, leading to a preferential selection of low-affinity cells to the memory compartment. Low-affinity memory may be at a disadvantage in mounting a protective immune response since immune-pathogen recognition is largely determined by the binding affinity between an immune receptor and antigenic epitopes. On the other hand, immune-pathogen recognition is cross-reactive, which would allow memory receptors to recognize slightly evolved forms of the antigen, in response to which they were originally generated.

We propose that the program for differentiation of immune cells to memory should be viewed in light of the immune system's coevolution with pathogens. We have developed a theoretical framework that incorporates the kinetics and energetics of memory responses as ingredients of memory strategy, which we seek to optimize under various evolutionary scenarios. We propose that the hardwired affinity-dependent regulatory measures for memory differentiation could be understood as a way to optimize the long-term utility of immune memory against evolving pathogens. Individuals encounter many distinct pathogens with varying evolutionary rates, ranging from relatively conserved pathogens like chickenpox to rapidly evolving viruses like influenza. To battle such a spectrum of evolving pathogens, we propose that an optimal immune system should store a combination of low-affinity memory with high cross-reactivity to counter evolving pathogens, and high-affinity and specific memory to counter the relatively conserved pathogens—a strategy consistent with B-cell memory, which often involves storage of both cross-reactive IgM and high-affinity IgG receptors (*Shlomchik, 2018*; *McHeyzer-Williams et al., 2018*). Lastly, we study the impact of organisms' life expectancy on their evolved memory strategies and predict that cross-reactive memory should dominate the immune response in short-lived organisms that encounter only a few pathogens.

Previous work on theoretical modeling of cellular differentiation together with experiments has been instrumental in understanding immune memory generation; for example see reviewed work in *Perelson and Weisbuch, 1997*; *Altan-Bonnet et al., 2020*. For example, mechanistic models have indicated the importance of signal integration at the cellular level (*Laffleur et al., 2014*) and the relevance of stochastic effects at the population level (*Hawkins et al., 2007*), to explain heterogeneous cell fate decisions for the generation of memory. Our statistical framework aims to characterize high-level features for an optimal memory strategy, without relying on mechanistic details of the underlying process, some of which are at least partially unknown (*Bialek, 2012*; *Nourmohammad et al., 2013*). In the case of the immune system, statistical models have provided an intuition for how an immune repertoire should be organized to optimally counter diverse pathogens (*Perelson and Oster, 1979*; *Mayer et al., 2015*; *Bradde et al., 2020*). In a similar fashion, optimal memory strategies identified by our model provide a baseline to gauge the performance of real immune systems in storing and utilizing memory.

## Model

The efficacy of an immune response to a pathogen is determined by two key factors: (i) the affinity of immune-pathogen recognition (i.e. energetics) and (ii) the speed of response (i.e. kinetics) to neutralize an infection.

Recognition of a pathogen (or its antigenic epitope) $\upsilon$ by an immune receptor $r$ is mediated by the affinity of the molecular interactions $E(r, \upsilon)$ between them. We describe cross-reactive immune-pathogen recognition in an immune *shape space* (*Perelson and Oster, 1979*), where receptors located near each other in shape space can recognize similar antigens, and in the complementary space, antigens that are close to each other can be recognized by the same immune receptor (*Figure 1*). We express the binding affinity between a receptor $r$ and an arbitrary target antigen $\upsilon$ in terms of the antigenic distance $d_r(\upsilon) = \|\upsilon - \upsilon_r^*\|$ between the receptor's cognate antigen $\upsilon_r^*$ and the target $\upsilon$: $E(r, \upsilon) \equiv E(d_r(\upsilon))$.

Physico-chemical constraints in protein structures can introduce a tradeoff between immune receptors' affinity and cross-reactivity. Although we lack a systematic understanding of these structural constraints, affinity-specificity tradeoffs have been reported repeatedly for B-cells and antibodies (*Wedemayer et al., 1997*; *Frank, 2002*; *Li et al., 2003*; *Wu et al., 2017*; *Mishra and Mariuzza, 2018*; *Fernández-Quintero et al., 2020*). Specifically, while affinity maturation can significantly

increase the binding affinity of a B-cell receptor, it also makes the receptor more rigid and specific to its cognate antigen (*Wedemayer et al., 1997*; *Li et al., 2003*; *Mishra and Mariuzza, 2018*; *Fernández-Quintero et al., 2020*). Broadly neutralizing antibodies (bNAbs) appear to be an exception to this rule since they have high potency and can react to a broad range of viral strains. However, it should be noted that bNAbs often react to vulnerable regions of a virus where escape mutations are very deleterious, including the CD4 binding site of HIV or the stem proteins in influenza (*Mascola and Haynes, 2013*; *Lee and Wilson, 2015*). In other words, the majority of bNAbs are not cross-reactive per se, but they are exceptionally successful in targeting conserved epitopes in otherwise diverse viral strains.

To qualitatively capture this affinity-specificity tradeoff, we use a simple functional form: We assume that the binding affinity of a receptor $r$ to an antigen $v$ depends on the antigenic distance $d_r(v)$ through a kernel with a specificity factor $\alpha$ and a shape factor $\theta$ such that, $E(r, v) \equiv E_{\alpha,\theta}(d_r(v)) \sim \alpha \exp[-(\alpha d_r(v))^{\theta}]$, with $\theta \geq 0$. This affinity function defines a receptor's binding profile over the space of antigens. As specificity $\alpha$ increases (or cross-reactivity $1/\alpha$ decays), the binding affinity profile sharpens and binding becomes more restrictive to antigens closer to the receptor's cognate antigen (*Figure 1*). Moreover, the absolute strength of binding to the cognate antigen (i.e. a receptor's maximum affinity) increases with specificity $\alpha$, resulting in a tradeoff between affinity and cross-reactivity. The parameter $\theta$ tunes the shape of the receptor's binding profile $E_{\alpha,\theta}(d_r(v))$, resulting in a flat function (i.e. no tradeoff) for $\theta = 0$, a double-sided exponential function for $\theta = 1$, a Gaussian (bell-curve) function for $\theta = 2$, and top-hat functions for $\theta \gg 2$; see Materials and methods.

Upon encountering a pathogen, the adaptive immune system mounts a response by activating the naïve repertoire (i.e. a novel response) and/or by triggering previously stored immune receptors in the memory compartment. A memory receptor often shows a reduced affinity in interacting with an evolved form of the pathogen. Nonetheless, memory plays a central role in protecting against re-infections since even a suboptimal memory can be kinetically more efficient than a naïve response, both in B-cells (*Tangye and Hodgkin, 2004*) and T-cells (*Whitmire et al., 2008*; *Martin et al., 2012*). Specifically, following an infection, memory B-cells initiate cell division about $1 - 2$ days earlier, and they are recruited to proliferate in $2 - 3$ times larger numbers compared to the naïve population (*Tangye et al., 2003*; *Tangye and Hodgkin, 2004*; *Blanchard-Rohner et al., 2009*). Once recruited, however, memory and naive cells have approximately a similar doubling time of about $t_{1/2} \approx 0.5 - 2$ days (*Tangye et al., 2003*; *Macallan et al., 2005*). Taken together, we can define an effective deliberation time $\tau \approx 1.5 - 5$ days for the naive population to reach an activity level (i.e. a clone size) comparable to the memory; see Materials and methods and *Figure 1*.

The decision to mount a naïve or a memory response depends on the energetics and the kinetics of the immune machinery, including the cross-reactivity of memory to recognize evolved pathogens and the deliberation time to mount a naive response upon infection—we refer to these choices as *memory strategies*. We expect that the biochemical machinery involved in making this decision upon an infection has been fine-tuned and selected over evolutionary time scales in order to utilize immune memory and mount an effective response against recurring pathogens. The theory of decision-making (*von Neumann and Morgenstern, 1944*; *Ortega and Braun, 2013*) enables us to characterize the response of the immune system as a rational decision-maker that chooses between two possible actions $a \in \{\text{naive}, \text{memory}\}$ each contributing a *utility* $U_a$ (Materials and methods). Specifically, the action of a rational decision-maker should follow an optimal distribution $Q_a$, which maximizes the expected utility while satisfying the constraints in processing new information, for example due to prior preferences (*von Neumann and Morgenstern, 1944*; *Ortega and Braun, 2013*). We assume that the immune system has no intrinsic prior for mounting a naive or a memory response against a given pathogen. In this case, the utility $U_a$ of an action (memory vs. naive) determines the type of response, and rational decisions follow a maximum entropy distribution $Q_a \sim \exp[\beta U_a]$ (*Jaynes, 1957*), where $\beta$ is the efficacy of information processing (see Materials and methods). As $\beta$ increases, a rational decision-maker more readily chooses the action with the highest utility. The expected utility of the immune response to an infection is equal to the sum of the utilities of a naive and a memory response, weighted by their respective probabilities: $\langle U \rangle = U_{\text{mem}} Q_{\text{mem.}} + U_{\text{naive}} Q_{\text{naive}}$. If memory is effective, the utility difference between mounting a memory or a naive response is

determined by the affinity of the interaction between the responding memory receptor $r_m$ and the infecting antigen $\upsilon$: $U_{\text{mem}} - U_{\text{naive}} = E_{\alpha,\theta}(r_m, \upsilon)$; see *Figure 1* and Materials and methods for details.

The time lag (deliberation) between memory and naive response also plays a key role in the decision-making process. On the one hand, if memory is inefficient, long deliberations would allow pathogens to proliferate, incurring a larger cost $\Omega_\tau$ to a host prior to activation of a novel response; this cost can be interpreted as the negative utility of naïve response $U_{\text{naive}} \equiv -\Omega_\tau$. On the other hand, a long deliberation would allow the immune system to exploit the utility of a usable memory (i.e. process information), even if the available memory has only a slight advantage over a responsive naive receptor (see Materials and methods). Indeed, for a responsive memory, the information processing factor $\beta$ is equal to accumulated pathogenic load $\Gamma_\tau$ during the deliberation period $\tau$, and thus, we refer to $\beta$ as the *deliberation factor*.

The expected binding profile of stored memory $\overline{E_{\alpha,\theta}^{(i)}(\upsilon)}$ after $i^{th}$ round of re-infection with an antigen $\upsilon_i$ can be characterized as the superposition of the binding profiles following a memory or a naive response, weighted by the respective probability of each of these events (*Figure 1* and Materials and methods). Since mounting a sub-optimal memory against evolved variants of a reinfecting pathogen can still be kinetically favorable, the expected profile can deviate from the optimal profile of the cognate receptor centered around the infecting pathogen $E_{\alpha,\theta}(r_{\upsilon_i}, \upsilon)$ (*Figure 1*). This tradeoff between the kinetics and the energetics of immune response results in a *non-equilibrium decision-making Grau-Moya et al., 2018* by the immune system (Materials and methods). In analogy to non-equilibrium thermodynamics, we express this deviation as a dissipative cost of memory response $K_{\text{diss}}(t_i; \alpha, \theta)$ at the $i^{th}$ round of re-infection (time point $t_i$), which we quantify by the Kullback-Leibler distance between the expected and the optimal binding profiles $D_{KL}\left(\overline{E_{\alpha,\theta}^{(i)}(\upsilon)} || E_{\alpha,\theta}(r_{\upsilon_i}, \upsilon)\right)$, in units of the deliberation factor $\beta$ (*Figure 1*),

$$
\begin{aligned}
K_{\text{diss}}(t_i) &= \frac{1}{\beta} D_{KL}\left(\overline{E_{\alpha,\theta}^{(i)}(\upsilon)} || E_{\alpha,\theta}(r_{\upsilon_i}, \upsilon)\right) \\
&= \frac{1}{\beta} \sum_{\text{antigens:}\upsilon} \overline{E_{\alpha,\theta}^{(i)}(\upsilon)} \log\left[\frac{\overline{E_{\alpha,\theta}^{(i)}(\upsilon)}}{E_{\alpha,\theta}(r_{\upsilon_i}, \upsilon)}\right].
\end{aligned}
\tag{1}
$$

An optimal memory strategy should be chosen such that it maximizes the expected utility of the immune response $\langle U \rangle$, while minimizing the dissipation cost due to the non-equilibrium response $K_{\text{diss}}$, over the lifetime of an organism. To infer an optimal strategy, we introduce net utility that accounts for the tradeoff between the expected utility and dissipation at a given round of infection at time point $t_i$,

$$
U_{\text{net}}(t_i) = \langle U(t_i) \rangle - K_{\text{diss}}(t_i)
\tag{2}
$$

We infer the optimal memory protocol (i.e. the optimal memory specificity $\alpha^*$ and deliberation factor $\beta^*$) by maximizing the total net utility of memory responses throughout the lifetime of an organism (*Figure 1*),

$$
(\alpha^*, \beta^*) = \underset{\alpha,\beta}{\text{argmax}} \sum_{i:\text{infections}} U_{\text{net}}(t_i).
\tag{3}
$$

## Results

Efficient immune memory balances specificity and speed. The extent of cross-reactivity and deliberation needed for the memory to react to pathogens should be set by the amount of pathogenic evolution and more specifically, the antigenic divergence $\hat{\delta} \equiv \sqrt{\langle \|\upsilon_i - \upsilon_{i-1}\|^2 \rangle}$ that a pathogen traces between two infections. An example of such antigenic divergence is shown in Fig. *Figure 1D* for 40 years of H3N2 Influenza evolution along it first (most variable) evolutionary dimension (*Bedford et al., 2014*). We set to find an optimal immune protocol (i.e. specificity $\alpha^*$ and deliberation $\beta^*$) by maximizing the net utility $U_{\text{net}}$ of an immune system (*Equation 3*) that is trained to counter pathogens with a given antigenic divergence $\hat{\delta}$; see Fig. *Figure 1D* and Materials and methods for details on the optimization procedure.

To battle slowly evolving pathogens ($\hat{\delta} \leq 20\%$) an optimal immune system stores highly specific memory receptors, with a specificity that approaches the upper bound $\alpha_{\max}$; see *Figure 2A* and *Figure 2—figure supplement 2*, *Figure 2—figure supplement 3*. Importantly, the dependency of optimal specificity on antigenic divergence is insensitive to the cost of deliberation $\Omega$ prior to mounting a naive response (*Figure 2A*), the shape factor $\theta$ for the specificity profile (*Figure 2—figure supplement 2*), and the specificity threshold $\alpha_{\max}$ (*Figure 2—figure supplement 3*). For relatively conserved pathogens ($\hat{\delta} \simeq 0$), the highly specific memory (with $\hat{\alpha}^* \equiv \alpha^*/\alpha_{\max} \simeq 1$) stored from a previous

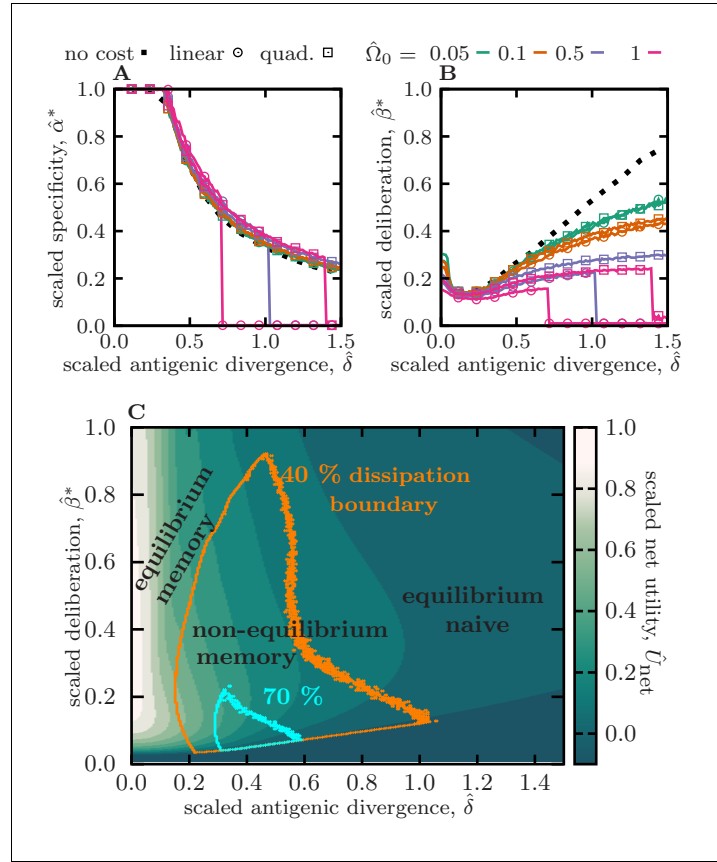

**Figure 2.** Optimal memory strategies against evolving pathogens. (A) and (B) show the optimal specificity $\hat{\alpha}^* \equiv \alpha^*/\alpha_{\max}$ and deliberation factor $\hat{\beta}^* \equiv \beta^*/\beta_{\max}$, scaled by their respective upper bounds, as a function of the antigenic divergence per infection, scaled by the cross-reactive range (or inverse of maximum specificity) $\hat{\delta} \equiv \delta/(\alpha_{\max}^{-1})$. Colors/markers indicate different naïve cost functions for deliberation, including no-cost $\hat{\Omega} \equiv \Omega/E_{\max} = 0$, linear cost $\hat{\Omega} = \hat{\Omega}_0 \hat{\beta}$, and quadratic cost $\hat{\Omega} = \hat{\Omega}_0 \hat{\beta}^2$, with varying amplitudes $\Omega_0$. (C) The heat map shows the expected rescaled net utility $\hat{U}_{\mathrm{net}} = U_{\mathrm{net}}/E_{\max}$ (*Equation 2*) per round of infection for an immune system with an optimal specificity $\hat{\alpha}^*$, as a function of rescaled antigenic divergence $\hat{\delta}$ and deliberation factor $\hat{\beta}$. Rescaling by $E_{\max}$ sets the magnitude of net utility to one, for a response to conserved antigens (with $\hat{\delta} = 0$) and in the limit of zero deliberation cost $\Omega \to 0$. Boundaries indicate different levels of dissipation, with orange and blue encompassing regions of $\geq 40\%$ and $\geq 70\%$ of the maximum dissipation $K_{\max}$, respectively. The three modes of immune response are indicate based on the magnitude of dissipation and net utility in each reagion: (i) equilibrium memory, (ii) non-equilibrium memory, and (iii) equilibrium naive. Simulation parameters, (A–C): $\alpha_{\max} = 4$, $\beta_{\max} = 10$, and $\theta = 2$, (C): linear deliberation cost function $\hat{\Omega} = \hat{\Omega}_0 \hat{\beta}$ with $\hat{\Omega}_0 = 0.1$. Results for other shape parameters $\theta$ and specificity thresholds $\alpha_{\max}$ are shown in *Figure 2—figure supplement 2*, *Figure 2—figure supplement 3*, respectively.

The online version of this article includes the following figure supplement(s) for figure 2:

**Figure supplement 1.** Utility, dissipation, and usage of optimal memory.
**Figure supplement 2.** Optimal memory strategies for different specificity shape factors $\theta$.
**Figure supplement 3.** Optimal memory strategies for different specificity thresholds $\alpha_{\max}$.

infection still has high affinity and remains centered and close to the reinfecting pathogens. Therefore, the immune system maintains a moderate level of deliberation to exploit this efficient memory during infections. However, as antigenic divergence grows, specific memory becomes less effective against future infections and therefore, the immune system reduces the deliberation factor to allow a timely novel response, once memory becomes inefficient (*Figure 2B*, *Figure 2—figure supplement 2*, *Figure 2—figure supplement 3*). The magnitude of deliberation decays as the cost of deliberation $\Omega$ increases but its overall dependency on antigenic divergence remains comparable for different cost functions (shown in *Figure 2B* for zero cost, and cost functions that grow linearly and quadratically with deliberation factor β). Overall, the net utility of the stored memory in response to slowly evolving pathogens is high (*Figure 2C*, *Figure 2—figure supplement 1*, *Figure 2—figure supplement 2*, *Figure 2—figure supplement 3*), while its dissipation remains small $K_{\mathrm{diss}} \simeq 0$ (*Figure 2C*, *Figure 2—figure supplement 1*, *Figure 2—figure supplement 2*, *Figure 2—figure supplement 3*). Therefore, in analogy to thermodynamics, we term this immune strategy with low dissipation as *equilibrium memory response*; *Figure 2C*.

To battle moderately evolving pathogens (with $\hat{\delta} \simeq 20\% - 60\%$), an optimal immune system stores cross-reactive memory (i.e. with a lower specificity $\hat{\alpha}$) that can recognize moderately evolved form of the primary antigen (*Figure 2A*, *Figure 2—figure supplement 2*, *Figure 2—figure supplement 3*). However, cross-reactive receptors tend to have lower affinities (*Wedemayer et al., 1997*; *Frank, 2002*), which could lead to deficient responses against antigens. Importantly, activation of energetically sub-optimal yet cross-reactive memory could be detrimental as it may hinder a stronger novel response without providing protective immunity to the host—a deficiency known as the original antigenic sin (*Francis, 1960*; *Vatti et al., 2017*). An optimal immune system can mitigate this problem by using kinetic optimization to tune the deliberation factor β in order to avoid an elongated memory engagement prior to a naive response. This optimization results in a smaller deliberation factor β (i.e. a faster naive response) compared to the scenario with slowly evolving pathogens, yet a long enough deliberation to allow the energetically suboptimal memory to react to an infection, whenever feasible (*Figure 2B*, *Figure 2—figure supplement 2*, *Figure 2—figure supplement 3*). With this kinetic optimization, the immune system can utilize cross-reactive memories through multiple rounds of infection (*Figure 2—figure supplement 1C*), yet with a declining efficiency and net utility as pathogens evolve away from the primary infection (*Figure 2C*, *Figure 2—figure supplement 1*, *Figure 2—figure supplement 2*, *Figure 2—figure supplement 3*). The prominent memory response to moderately evolving pathogens is dissipative with $K_{\mathrm{diss}} \gg 0$ (*Figure 2C*, *Figure 2—figure supplement 1*, *Figure 2—figure supplement 2*, *Figure 2—figure supplement 3*), and in analogy with thermodynamics, we term this dissipative immune strategy as *non-equilibrium memory response*; *Figure 2C*.

For extremely rapidly evolving pathogens ($\hat{\delta} > 60\%$), the immune system would not be able to store an efficient memory to battle future encounters, and hence, each infection would trigger a novel naive response — the reduced net utility of memory and the decay of memory usage in this regime are shown in *Figure 2C*, *Figure 2—figure supplement 1*, *Figure 2—figure supplement 2*, *Figure 2—figure supplement 3*, respectively. Without a protective memory, a novel response is triggered to counter each infection and it maturates specifically around the infecting pathogen, resulting in a non-dissipative naive-dominated immune response with $K_{\mathrm{diss}} \simeq 0$, which we term *equilibrium naive response*; *Figure 2C*.

It should be noted that when the cost of deliberation $\Omega$ is very high, utilizing memory against pathogens with relatively high evolutionary rates becomes highly unfavorable. In this extreme case, the immune system switches into a state where it invariably mounts a novel response upon an infection (*Figure 2—figure supplement 1C*), and it assures that memory is not utilized by setting the parameters for specificity α and deliberation β to zero (*Figure 2A,B*).

Our analyses in *Figure 2* indicate that a rational decision to become a memory or a plasma cell during an immune response should depend on the affinity of a cell's receptors and it should not be a stochastic choice with a constant rate throughout affinity maturation. Indeed, cell fate decision for B-cells during affinity maturation is highly regulated and dependent on receptors' affinity (*Good-Jacobson and Shlomchik, 2010*; *Kometani et al., 2013*; *Shinnakasu et al., 2016*; *Weisel et al., 2016*; *Shinnakasu and Kurosaki, 2017*; *Shlomchik et al., 2019*). Recent experiments have demonstrated that memory generation is highly correlated with the activity of the transcription factor

*Bach2* whose expression level is negatively regulated with the abundance of helper CD4+ T-cells (*Kometani et al., 2013*; *Shinnakasu et al., 2016*; *Shinnakasu and Kurosaki, 2017*). As the affinity of B-cell receptors increases during affinity maturation, more CD4+ T-cells are recruited to germinal centers, resulting in suppression of *Bach2* and a hence, a decline in production of memory cells (*Kometani et al., 2013*; *Shinnakasu et al., 2016*; *Shinnakasu and Kurosaki, 2017*). In other words, our adaptive immune system has encoded a negative feedback mechanism to store memory with intermediate affinity and cross-reactivity to suppress the production of highly specific memory, which is likely to be impotent against evolved pathogens in future infections.

## A mixture memory strategy is necessary to counter pathogens with a broad range of evolutionary rates

The decision to trigger an equilibrium or a non-equilibrium memory response depends on the extent of antigenic divergence that an immune system is trained to cope with (*Figure 2*, *Figure 2—figure supplement 1*, *Figure 2—figure supplement 2*, *Figure 2—figure supplement 3*). Equilibrium memory is highly effective (i.e. it has high net utility) against relatively conserved pathogens, however, it fails to counter evolving pathogens (*Figure 2C*). On the other hand, cross-reactive non-equilibrium memory is more versatile and can counter a broader range of evolved pathogens but at a cost of reduced net utility in immune response; *Figure 2C*, *Figure 2—figure supplement 1*, *Figure 2—figure supplement 2*, *Figure 2—figure supplement 3*.

An optimal immune system should have memory strategies to counter pathogens with varying evolutionary rates, ranging from relatively conserved pathogens like chickenpox to rapidly evolving viruses like influenza. We use our optimization protocol to find such memory strategies that maximize the net utility of an immune system that encounters evolving pathogens with (scaled) antigenic divergences uniformly drawn from a broad range of $\hat{\delta} \in [0\ 1.6]$; see Materials and methods. This optimization results in a bimodal distribution of optimal specificity for functional memory receptors $P(\alpha)$, with separated peaks corresponding to equilibrium ($\hat{\alpha} \sim 1$) and non-equilibrium ($\hat{\alpha} \sim 0.5$) memory (*Figure 3*, *Figure 3—figure supplement 1*). This result suggests that specific and cross-reactive memory

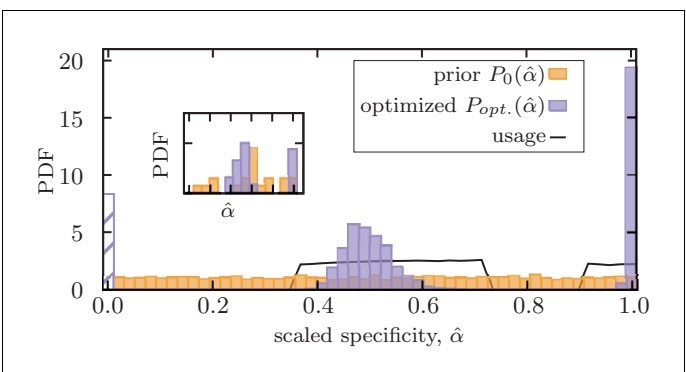

**Figure 3.** Mixed memory strategy against a mixture of pathogens with a broad range of evolutionary rates. Distribution of scaled optimized specificities $\hat{\alpha}^*$ for functional memory (purple) is shown for an immune system with a fixed deliberation factor $\hat{\beta} = 0.2$. A mixture strategy with a bimodal distribution of specificities $P(\hat{\alpha})$ is established to counter pathogens with a broad range of antigenic divergences. The dashed bar indicates stored memory with specificity $\alpha = 0$, which is not further used in response to infections. The solid line indicates the probability $P_{\text{usage}}$ that a stored memory with a given specificity is utilized in future infections (Materials and methods). Optimization is done by maximizing the net utility of immune response averaged over encounters with 1000 independently evolving antigens with (scaled) antigenic divergences drawn uniformly from a range $\hat{\delta} \in (0, 1.6)$ (Materials and methods). The distribution shows the ensemble statistics of functional memory accumulated from 200 independent optimizations, each starting from a flat prior for specificities (orange). The insert shows the optimized mixture strategy for one optimization with 3000 steps. Simulation parameters: $\alpha_{\text{max}} = 4$, $\beta_{\text{max}} = 10$, and $\theta = 2$.

The online version of this article includes the following figure supplement(s) for figure 3:

**Figure supplement 1.** Mixed memory strategy against pathogens for different deliberation factors $\hat{\beta}$.

strategies are complementary modes of immune response that cannot substitute each other. Moreover, non-equilibrium memory tends to be flexible and moderate values of cross-reactivity $1/\hat{\alpha}$ can counter a range of antigenic divergences, without a need for fine-tuning. Therefore, upon production of memory, an optimal immune system should harvest both specific equilibrium memory and cross-reactive non-equilibrium memory, as it does not have a priori knowledge about the evolutionary rate of the infecting pathogen.

Interestingly, the adaptive immune system stores a mixture of IgM and class-switched IgG isotypes of B-cell memory that show different levels of specificity. IgM memory is an earlier product of affinity maturation with higher cross-reactivity and a lower affinity to antigens, reflecting a non-equilibrium memory that can counter evolving pathogens. On the other hand, memory from class-switched (e.g. IgG) isotype is produced during later stages of affinity maturation and is highly specific to the infecting pathogen, reflecting equilibrium memory that is effective against relatively conserved pathogens (*Weisel et al., 2016*). Storing a mixture of IgM and class-switched IgG memory is consistent with our recipe for optimal immune strategies to counter pathogens with a broad range of evolutionary rates.

## Cross-reactive memory dominates immune response in organisms that encounter fewer pathogens over a shorter lifetime

So far, our analysis has focused on maximizing the net utility of immune response, assuming that organisms encounter many such infections throughout their lifetime. This optimization provides a recipe for optimal immune strategies in response to commonly infecting pathogens. However, the expected frequency of infections is also an important factor that can inform immune strategies. For example, imagine the extreme case that an immune system expects to encounter a pathogen at most only once during an organism's lifetime, for example in short-lived organisms. In this case, there is no benefit in keeping a memory even to counter extremely conserved pathogens, for which memory would be otherwise very beneficial.

To study the impact of infection frequency on immune strategies, we use our optimization procedure to maximize the net utility of immune response, while setting a bound on the number of infections throughout an organism's lifetime (see Materials and methods). Organisms with an *unrealistically* very short lifetime (measured in units of the number of infections) experience only a few infections, and therefore, a small (cumulative) antigenic drift from the primary infection during their lifetime $\hat{\delta}\sqrt{\text{lifetime.}} \lesssim 1$. In this case, it would be sufficient for an optimal immune system to generate specific memory ($\hat{\alpha} \approx 1$), which can mount an effective response with only an intermediate deliberation ($\hat{\beta} \sim 0.4$) upon reinfection (*Figure 4A–B*), even for pathogens with a moderate evolutionary rate (*Figure 4B*). Organisms with moderately short lifetime experience evolutionary divergence of reinfecting antigens. In this regime, the immune system stores cross-reactive memory (smaller $\hat{\alpha}$) and uses a larger deliberation factor $\hat{\beta}$ such that this lower-affinity and often off-centered memory can mount an effective response to evolved infections (*Figure 4A–B*). Since the organism is relatively short-lived, such cross-reactive memory could be sufficient throughout the whole lifetime of the organism, without a need for renewal.

Organisms with long lifetimes, with pathogen encounters that surpassing the threshold $c^*$, expect higher re-infections with pathogens that are highly diverged from the primary infection. In this case, an optimal immune strategy switches from storing and utilizing cross-reactive memory to generating more specific memory receptors (*Figure 4A*). This specific memory would not hinder activation of preventive novel responses against evolved pathogens (the problem known as original antigenic sin), resulting in continual renewal of memory during organisms' lifetime. In this regime, the deliberation factor also decreases to facilitate novel responses against antigens that are not readily recognized by memory (*Figure 4A–B*). The increase in memory specificity from short- to long-lived organisms is more substantial for immune strategies optimized to counter relatively conserved pathogens, that is the specific equilibrium memory (*Figure 2C*, *Figure 4A*), compared to the memory against evolving pathogens, that is the cross-reactive non-equilibrium memory (*Figure 2C*, *Figure 4B*). The exact value of the transition threshold $c^*$ depends on the expected antigenic divergence $\delta$ during pathogenic evolution and the details of the immune machinery, and specifically the cost of deliberation $\Omega(\tau)$ due to an elevated level of pathogenic proliferation prior to a novel response (*Figure 4—*

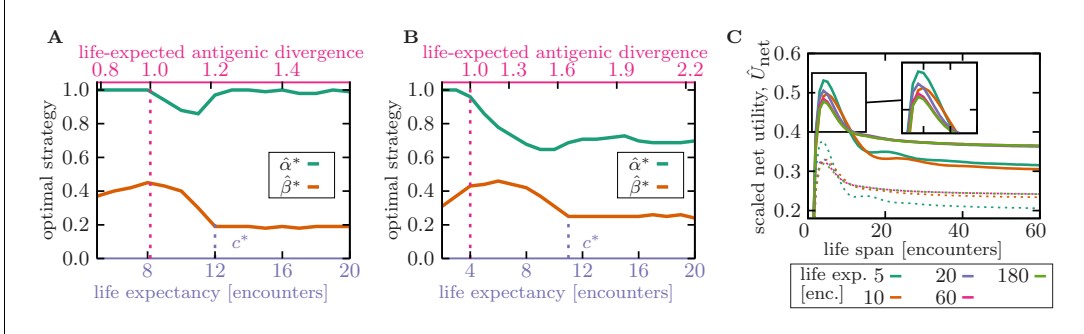

**Figure 4.** Life expectancy influences the specificity of optimal memory. (A,B) Memory strategies, that is, optimal rescaled specificity $\hat{\alpha}^*$ (green) and deliberation factor $\hat{\beta}^*$ (orange) are shown as a function of the organism's life expectancy (bottom axis) and the corresponding expected antigenic divergence over the organism's life-time $\hat{\delta}\sqrt{\text{lifetime}}$ (top axis). Antigenic divergence (per encounter) of the infecting pathogen is $\hat{\delta} = 0.35$ in (A) and $\hat{\delta} = 0.5$ in (B). Memory is highly specific in organisms with very short lifetimes, during which re-infections with evolved forms of a pathogen are unlikely (i.e. when life-expected antigenic divergence is smaller than 1, indicated by a dotted pink line). Memory becomes more cross-reactive with a smaller deliberation in organisms with (realistic) short lifetimes, up to a transition point $c^*$ (indicated by dotted purple line), after which specificity increases again. (C) Scaled net utility $\hat{U}_{\text{net}}$ is shown as a function of organism's life span, whose immune strategies ($\hat{\alpha}^*$, $\hat{\beta}^*$) are optimized for a specified life expectancy (colors as indicated in the legend). Net utility for memory optimized against pathogens with antigenic divergence $\hat{\delta} = 0.35$ (panel A) and $\hat{\delta} = 0.5$ (panel B) are shown by full and dashed lines, respectively. Life span and life expectancy are measured in units of the number of pathogenic encounters during lifetime. Simulation parameters: linear deliberation cost function $\Omega = \Omega_0 \hat{\beta}$ with an amplitude $\hat{\Omega}_0 = 0.1$, $\alpha_{\max} = 4$, $\beta_{\max} = 10$, and $\theta = 2$.

The online version of this article includes the following figure supplement(s) for figure 4:

**Figure supplement 1.** Pathogen encounter threshold to transition between cross-reactive and specific memory.

*figure supplement 1*). However, the qualitative trend for cross-reactivity as a function of the organism's lifetime remain consistent across a range of parameters.

The results in *Figure 4* predict that organisms with few pathogenic encounters or a shorter lifespan should generate more cross-reactive and lower affinity (i.e. a naive-type) memory receptors. Indeed, consistent with our prediction, analysis of immune repertoire data indicates that sequence features of memory and naïve B-cell receptors tend to be more similar to each other in mouse compared to humans that enjoy a longer life expectancy (*Sethna et al., 2017*). Nonetheless, more comprehensive data on cross-species comparison of immune strategies is needed to test our predictions.

With the increase in human life expectancy, a pressing question is how well our immune system could cope with a larger number of pathogenic challenges that we are now encountering throughout our lifetimes? Aging has many implications for our immune machinery and the history of infections throughout lifetime leaves a complex mark on immune memory that can have long-lasting consequences (*Saule et al., 2006*), which has also been studied through theoretical modeling (*Mayer et al., 2019*). In our framework, we can study one aspect of this problem and ask how an immune strategy optimized to battle a given number of infections would perform if the organism were to live longer or equivalently, to encounter pathogens more frequently. *Figure 4C* shows that cross-reactive memory generated by an immune system optimized to counter few infections (short life expectancy) becomes highly inefficient (i.e., with a lower net utility $U_{\text{net}}$) as the number of encounters increases beyond the organism's expectation (long life span)—an effect that may be in part responsible for the observed decline in the efficacy of our adaptive immunity as we age.

## Discussion

Memory is central to our adaptive immunity by providing a robust and preventive response to reinfecting pathogens. In the presence of continually evolving pathogens, immune memory is only beneficial if receptors can recognize evolved antigens by cross-reactivity. However, biophysical constraints can impose a trade-off between affinity and cross-reactivity of antibodies. Specifically, as receptors undergo affinity maturation, their structures become more rigid and less cross-reactive,

while affinity increases (*Wedemayer et al., 1997*; *Frank, 2002*; *Li et al., 2003*; *Wu et al., 2017*; *Mishra and Mariuzza, 2018*; *Fernández-Quintero et al., 2020*). Consistent with recent experiments (*Weisel et al., 2016*; *Shinnakasu et al., 2016*; *Recaldin and Fear, 2016*; *Shinnakasu and Kurousaki, 2017*; *Viant et al., 2020*), we show that memory differentiation should be regulated to preferentially produce lower affinity receptors, which can allow cross-reactive recognition of evolved pathogens. To overcome the resulting energetic impediment of these memory receptors, we infer that the immune system should tune the kinetics of the immune response and allocate a longer deliberation time for memory to react before initiating a novel response—a feature that is also in accordance with observations (*Tangye et al., 2003*; *Tangye and Hodgkin, 2004*; *Blanchard-Rohner et al., 2009*). Co-optimizing kinetics and energetics of memory ensures an effective response against evolving pathogens, throughout an organism's lifetime.

Optimal cross-reactive immune memory provides a long-term advantage to an organism, yet it may seem energetically sub-optimal over short time scales (*Figure 1*). One important consequence of a sub-optimal memory response is known as original antigenic sin, where cross-reactive memory from primary infections could interfere with and suppress a protective novel response (*Francis, 1960*; *Vatti et al., 2017*). The viral exposure history and the original antigenic sin may have profound consequences on protective immunity against evolving viruses (*Cobey and Hensley, 2017*). For example, the 2009 H1N1 pandemic triggered memory responses in individuals with childhood exposures to seasonal H1N1 (*Linderman and Hensley, 2016*; *Li et al., 2013*; *Hensley, 2014*), which in some led to a highly focused antibody response toward the conserved epitopes of H1N1. This focus was a problem when in 2013–2014 the pandemic H1N1 acquired mutations in those epitopes (*Linderman and Hensley, 2016*), resulting in a disproportionate impact of infection on middle-aged individuals with pre-existing memory (*Petrie et al., 2016*). This recent example, among others, showcases how immune history and antigenic sin can impact a population's immune response to the a rapidly evolving virus like influenza.

Composition of the immune memory coupled with the exposure history of the host should be taken into account when designing new vaccines (*Cobey and Hensley, 2017*). For example, current vaccine strategies against influenza use sera isolated from ferrets infected with the virus to measure the antigenic distance of circulating strains against the previous years (*Smith et al., 2004*). However, these ferrets have no immune history for influenza and the antibodies they produce may be distinct from the immune response in the adult population with prior memory, resulting in incorrect measures of antigenic distances (*Hensley, 2014*). This problem has been recognized by the World Health Organization and there is now an effort to choose vaccine strains based on human serology.

The impact of immune deficiency related to the original antigenic sin can even be more pronounced due to changes in an organism's life expectancy. Importantly, we show that immune strategies optimized to benefit short-lived organisms produce highly cross-reactive memory (*Figure 4*). If an organism's life-expectancy increases, which is the case for humans, it would be likely for individuals to encounter evolved forms of a pathogen at antigenic distances larger than expected by their immune systems. In this case, cross-reactive memory, optimized for a shorter lifetime, could still be activated but with lower efficacy, which could suppress a protective novel response, consistent with original antigenic sin. It is therefore important to consider sub-optimality of immune strategies in the face of extensive elongation of the human lifespan as one of the plausible factors responsible for immune deficiencies brought by aging.

One characteristic of memory B-cells, which is currently missing from our model, is their ability to seed secondary germinal centers and undergo further affinity maturation upon reinfection. Evolvability of memory B-cells can allow cross-reactive memory to further specialize against evolved pathogens, without a need to start a germinal center reaction from an un-mutated naive receptor. Interestingly, different experiments suggest that the capacity of memory to re-diversify depends on various factors including the memory isotype (IgM vs. class-switch receptors), the type of antigenic target (viruses vs. others) and the extent of memory maturation (*Shlomchik, 2018*; *McHeyzer-Williams et al., 2018*). Therefore, it is interesting to extend our model to study how evolvability of memory can influence its longterm utility to respond to evolving pathogens, and especially viruses.

Evolvability of memory is also relevant for characterizing the dynamics of immune response to chronic viral infections like HIV. Analyses of immune repertoires in HIV patients over multiple years of infection have shown a rapid turnover and somatic evolution of B-cell clonal lineages to counter

the evolution of the virus within hosts (*Nourmohammad et al., 2019*). It would be interesting to see how the constant pressure from the evolving HIV on a host's immune system impacts the dynamics and efficacy of immune memory over time. In addition, understanding the limits of memory re-diversification is instrumental in designing successive vaccination protocols with antigen cocktails to drive extensive affinity maturation of BCR lineages to elicit broadly neutralizing antibodies *Wang et al., 2015*; *Shaffer et al., 2016*; *Stephenson et al., 2020*—an approach that is the current hope for universal vaccines against rapidly evolving viruses like HIV.

Although mechanistically distinct from B-cells, T-cells also differentiate into effector and memory in response to infections. The T-cell response does not involve affinity maturation by hypermutations. However, competition among T-cells with varying receptor affinities acts as selection that leads to immuno-dominant responses by the high-affinity clones. Receptor affinity and the subsequent T-cell signaling determine the extent of clonal expansion and differentiation to an effector versus a memory T-cell population (*Kim and Williams, 2010*). Although it is still unresolved as how T-cell signaling determines cell fate decision, the process is known to be highly regulated (*Rutishauser et al., 2009*; *Roychoudhuri et al., 2016*). Notably, the transcription factor IRF4 selectively promotes expansion and differentiation of high-affinity cytotoxic T-cells into effectors. In contrast, low-affinity T-cells are lost or they could differentiate into early memory (*Man et al., 2013*). There is also accumulating evidence for the circulation of cross-reactive memory T-cells, which often result in protective immunity against evolving forms of a virus (*Greenbaum et al., 2009*; *Sette and Crotty, 2020*), but could also be detrimental by suppressing novel and specific responses—an effect similar to the original antigenic sin by B-cells (*Selin et al., 2004*). Taken together, there are parallels between differentiation of T-cells and B-cells to memory, and it will be interesting to investigate the advantages of storing cross-reactive (and plausibly low-affinity) T-cell memory as a strategy to counter evolving pathogens.

## Materials and methods

All codes for simulations and numerical analysis can be found at: https://github.com/StatPhysBio/ImmuneMemoryDM (swh:1:rev:c71f7ab35ebcdd251e4a26fdf9628386fe404e86; *Schnaack, 2021*).

### Numerical optimization

Numerical optimization is performed on ensembles of immune systems that encounter evolving pathogens. Recognition of an evolved pathogen at the $i^{th}$ round of infection $v_i$ by a memory that was stored in response to a primary infection $v_0$ ($0^{th}$ round) depends on the antigenic distance $d_i = \|v_i - v_0\|$. We model pathogenic evolution as diffusion in the antigenic shape space. In this model, the expected antigenic distance between the primary infection $v_0$ and the evolved antigen $v_i$ can be characterized as, $\langle d_i^2 \rangle \equiv \langle \|v_i - v_0\|^2 \rangle = \zeta^2(t_i - t_0) = i\delta^2$, where $\zeta$ is the diffusion coefficient (i.e. the evolutionary rate) and $\delta$ is the (averaged) antigenic divergence per round of infection. Importantly, this relationship does not depend on the dimensionality of the antigenic shape space, which in general, is difficult to characterize. We simulate pathogenic evolution relative to a primary infection by drawing the corresponding antigenic distance $d_i$ of the $i^{th}$ round of infection from a normal distribution with mean $\delta\sqrt{i}$ and standard deviation $0.05\delta\sqrt{i}$. The width of this normal distribution characterizes the fluctuations in the mean divergence between infections and reflects how the evolutionary trajectory of a pathogen samples the multi-dimensional shape space surrounding the antigen from the primary infection. Nonetheless, our results are insensitive to the exact choice of this width.

To characterize optimal specificity $\alpha^*$ and deliberation factor $\beta^*$ (*Figure 2*, *Figure 3*, *Figure 4*), we simulate ensembles of immune systems with different immune strategies $(\alpha, \beta)$, chosen uniformly from the range $\alpha \in [0, \alpha_{\max}]$ and $\beta \in [0, \beta_{\max}]$, with 500 increments in both parameters. Each immune system experiences successive rounds of infection with an evolving pathogen with a given antigenic divergence $\delta$. During each encounter, the immune system chooses between utilizing an existing memory or initiating a novel response according to *Equation 6*. The net utility of each encounter is calculated according to *Equation 2*. We estimate the expected net utility per encounter over a lifetime of 60 total encounters and repeat this experiment across $10^5$ independent ensembles to find the optimal immune strategies $(\alpha^*, \beta^*)$ with the highest net utility. As shown in *Figure 4*, simulating

up to 60 encounters is sufficient for the inference of optimal strategies in the asymptotic regime (i.e. a long lifetime).

To characterize optimal immune strategies against a mixture of pathogens with distinct levels of antigenic divergences, we define the mixture immune strategy by a set of specificities $\vec{\alpha} = \{\alpha_i\} = (\text{with}, i = 1, \ldots, N_m)$, where each $\alpha_i$ is a degree of specificity that a stored memory receptor can potentially have, and $N_m$ is the number of possible specificity strategies that an immune system can choose from. The probability that an immune system with the mixture strategy $\vec{\alpha}$ recognizes a pathogen $\upsilon$ through a memory response follows from an extension of = *Equation 6*,

$$
\begin{aligned}
P_{\text{recog.}}^{(m)}(\vec{\alpha}, \upsilon) &= 1 - \prod_{\text{specificity}:\alpha_i} \left( 1 - P_{\text{recog.}}^{(m)}(r_m^{\alpha_i}, \upsilon) \right) \\
&= 1 - \prod_{\text{specificity}:\alpha_i} e^{-E_\theta(r_m^{\alpha_i}, \upsilon)\Gamma(\tau)} = 1 - e^{-\sum_{\alpha_i} E_\theta(r_m^{\alpha_i}, \upsilon)\Gamma(\tau)} \equiv 1 - e^{-\tilde{\beta}\overline{E}_\theta(\upsilon)}
\end{aligned}
\tag{4}
$$

where $\overline{E}_\theta(\upsilon) = \frac{1}{N_m}\sum_{r_m^\alpha} E_\theta(r_m^{\alpha_i}, \upsilon)$ is the expected affinity of memory (with distinct specificities) against antigen $\upsilon$ in an immune repertoire and $\tilde{\beta} \equiv N_m\beta$ is an effective deliberation factor for all choices of specificity. It should be noted that this effective deliberation factor $\tilde{\beta}$ is an extensive quantity with respect to the number of specificity strategies that an immune system can choose from, and therefore, is comparable across immune systems with different numbers of strategies.

We set out to characterize the mixture strategy as the probability $P_\beta(\alpha)$ based on which an immune system with a given effective deliberation factor $\tilde{\beta}$ should store a memory receptor with specificity $\alpha$, in order to optimally counter infecting pathogens with distinct antigenic divergences, drawn from a distribution $P(\delta)$. We start our optimization by defining a uniform mixture strategy, where the elements of the immune specificity vector $\vec{\alpha} = \{\alpha_i\}$ (of size $N_m = 20$), are drawn uniformly from the range $[0, \alpha_{\max}]$. Each optimization step aims to improve the specificity vector $\vec{\alpha}$ to maximize the net utility (per encounter) of the mixture immune response $U_{\text{net}}(\vec{\alpha}^k)$ against 1000 independently evolving antigens whose (scaled) antigenic divergences are drawn uniformly from the range $\hat{\delta} = [0, \hat{\delta}_{\max}]$. We use stochastic simulations to estimate the net utility of the mixture strategy $U_{\text{net}}(\vec{\alpha}^k)$, whereby the relative affinity of memory receptors (with varying specificities), $E_\theta(r_m^{\alpha_i}, \upsilon)/\overline{E}_\theta(\upsilon)$, determines the stochastic rate of their response to the infecting antigen $\upsilon$. The net utility (per encounter) of the immune response against each of the 1000 independently evolving antigens is estimated by averaging over a host's lifetime with 200 rounds of pathogenic encounters. We update the mixture strategy over 3000 steps, using local gradient ascent by sampling 100 points in the space of specificity vectors at each step to maximize net utility,

$$
\vec{\alpha}^{k+1} = \vec{\alpha}^k + \epsilon\nabla U_{\text{net}}(\vec{\alpha}^k)
\tag{5}
$$

Here, $k$ indicates the optimization step and $\epsilon = 0.1$ is a hyper-parameter for gradient ascent. We repeat the optimization process starting from 200 independently drawn initial uniform mixture strategies $\vec{\alpha}^0$ to characterize the ensemble of optimal memory strategies $P_\beta(\alpha)$ against pathogens with distinct antigenic divergences drawn uniformly from a given range $\hat{\delta} = [0, \hat{\delta}_{\max}]$, as shown in *Figure 3*. We also characterize the probability that a stored memory with a given specificity is utilized against future infections (solid line in *Figure 3*). To do so, we test the optimized ensemble of specificities $P_\beta(\alpha)$ against 5000 independent pathogens with antigenic divergences drawn uniformly from the range $\hat{\delta} = [0, \hat{\delta}_{\max}]$. We evaluate the usage of a memory with a given specificity $\alpha$ (solid line in *Figure 3*) as the conditional probability $P_\beta(\text{use}\,\alpha|\text{produce}\,\alpha)$ for using that memory given that it is produced (i.e. drawn from the distribution $P_\beta(\alpha)$).

## Model of evolutionary decision-making for adaptive immune response
### Kinetics of naive and memory immune response
Upon encountering a pathogen, the adaptive immune system mounts a response by activating the naïve repertoire (i.e. a novel response) and/or by triggering previously stored immune receptors in the memory compartment. A memory receptor often shows a reduced affinity in interacting with an evolved form of the pathogen. Nonetheless, memory plays a central role in protecting against re-infections since even a suboptimal memory can be kinetically more efficient than a naive response,

both in B-cells (*Tangye and Hodgkin, 2004*) and T-cells (*Whitmire et al., 2008*; *Martin et al., 2012*). First, memory cells are fast responders and initiate cell division about $\tau_0 \approx 1-2$ days before naive cells (*Tangye et al., 2003*; *Tangye and Hodgkin, 2004*; *Blanchard-Rohner et al., 2009*). Second, the number of memory cells that are recruited to proliferate and differentiate to effector cells is $b \approx 2-3$ times larger than the number of naive cells (*Tangye et al., 2003*; *Tangye and Hodgkin, 2004*). Once recruited, however, memory and naive cells have approximately a similar doubling time of about $t_{1/2} \approx 0.5-2$ days (*Tangye et al., 2003*; *Macallan et al., 2005*). Putting these kinetic factors together, we can define an effective deliberation time $\tau$ for the naive population to reach an activity level (i.e. a population size) comparable to the memory. Assuming an exponential growth during the early stages of memory and naïve proliferation, the deliberation time can be estimated in terms of the kinetic factors by $\tau = \tau_0 + t_{1/2} \ln b / \ln 2$ and it is within a range of $\tau \approx 1.5 - 5$ days; see *Figure 1*.

## Energetics of immune recognition

We assume that each immune receptor $r$ has a cognate antigen $v_r^*$ against which it has the highest affinity. We express the binding affinity between a receptor $r$ and an arbitrary target antigen $v$ in terms of the antigenic distance $d_r(v) = \|v - v_r^*\|$ between the receptor's cognate antigen $v_r^*$ and the target $v$: $E(r, v) \equiv E(d_r(v))$. This distance-dependent binding affinity is measured with respect to the affinity of unspecific antigen-receptor interactions, sufficient to trigger a generic naïve response.

Physico-chemical constraints in protein structures can introduce a tradeoff between immune receptors' affinity and cross-reactivity (i.e. ability to equally react to multiple targets). Prior to affinity maturation, the structure of naïve receptors is relatively flexible whereas hypermutations often reconfigure the active sites of a receptor and make them more specific so that they match their target antigens like a lock and key (*Wedemayer et al., 1997*; *Frank, 2002*). As a result, the IgM class of antibodies, which are the first line of defense in B-cell response, often have low affinities, yet they are cross-reactive and can recognize mutated forms of the same epitope. On the other hand, the high-affinity IgG class of antibodies, which are the late outcomes of affinity maturation in germinal centers, have higher affinities but bind very specifically to their cognate antigen (*Frank, 2002*). Broadly neutralizing antibodies (bNAbs) are exceptions to this rule since they often have high potency and can react to a broad range of viral strains. However, bNAbs often react to vulnerable regions of a virus where escape mutations are very deleterious (*Mascola and Haynes, 2013*). In other words, the majority of bNAbs are not cross-reactive per se, but they are exceptionally successful in targeting conserved epitopes in otherwise diverse viral strains. Nevertheless, an affinity-specificity tradeoff has been reported for a bNAb against the hemagglutinin epitope of influenza (*Wu et al., 2017*).

We use a simple functional form to qualitatively capture the tradeoff between cross-reactivity and affinity of antigen-receptor binding interactions: We assume that the binding affinity of a receptor $r$ to an antigen $v$ depends on the antigenic distance $d_r(v) = \|v - v_r^*\|$ through a kernel with a specificity factor $\alpha$ and a shape factor $\theta$ such that, $E(r, v) \equiv E_{\alpha,\theta}(d_r(v)) \sim \alpha \exp[-(\alpha\|v - v_r^*\|)^\theta]$, with $\theta \geq 0$. The width of this binding profile (i.e. the cross-reactivity) is set by the inverse of the specificity factor $1/\alpha$ (*Figure 1*), which decays as the height of the function (i.e. the maximum affinity) increases. The parameter $\theta$ tunes the shape of the receptor's binding profile $E_{\alpha,\theta}(d_r(v))$, resulting in a flat function (i.e. no tradeoff) for $\theta = 0$, a double-sided exponential function for $\theta = 1$, a Gaussian (bell-curve) function for $\theta = 2$, and top-hat functions for $\theta \gg 2$. Structural constraints and molecular features of protein receptors define a bound on the minimum cross-reactivity or equivalently, a maximum specificity $\alpha_{\max}$, achievable by a receptor. Using this bound, we define rescaled specificity $\hat{\alpha} \equiv \alpha/\alpha_{\max}$ to characterize the energetics of an immune response in a dimensionless form.

## Immune response to evolving pathogens

Upon primary infection (i.e. an encounter with a novel pathogen) naive immune receptors with moderate affinity are activated to develop a specific response through affinity maturation (*Figure 1*). Since the naive repertoire is diverse enough to contain receptors of moderate affinity against different antigens, we assume that the affinity of responsive naïve receptors, and hence, the strength of a primary immune response to be approximately the same for all pathogens. This simplification becomes less accurate as the immune system ages and the supply of effective receptors become more scarce.

Following a naive response to a primary infection and the subsequent affinity maturation, the immune system stores memory cells with an enhanced affinity to use them against future infections (*Janeway et al., 2005*; see *Figure 1*). Therefore, the cognate antigen $v_{r_m}^*$ for a given memory receptor $r_m$ is an epitope derived from the primary infection that led to the formation of memory, which we denote by $v_0$ with a subscript that indicates round of infection. Thus, the binding profile $E_{\alpha,\theta}(r_m, v)$ of the memory receptor $r_m$ is peaked around the primary antigenic epitope $v_{r_m}^* = v_0$ (*Figure 1*). As pathogens evolve globally to escape the immune challenge, drugs, or vaccination, they drift away from the primary antigen in antigenic space. We model this antigenic shift as a diffusion in shape space whereby a reinfecting pathogen at the $i^{th}$ round of infection $v_i$ is *on average* at a distance $\delta = \sqrt{\langle \|v_i - v_{i-1}\|^2 \rangle}$ from the previous infection $v_{i-1}$. This antigenic shift is proportional to the rate of pathogen evolution $\zeta_v$ and the average time between infections $\Delta t = t_i - t_{i-1}$, such that $\delta \propto \zeta_v \sqrt{\Delta t}$. A cross-reactive memory can mount a response to an evolved antigen, yet with a reduced affinity that decays with antigenic shift; see *Figure 1*. It should be noted that the minimum level of receptor's cross-reactivity (or maximum specificity) $(\alpha_{\max})^{-1}$ defines a natural scale against which we can measure antigenic divergence $\delta$ and hence, form a dimensionless measure of antigenic divergence $\hat{\delta} \equiv \delta / (\alpha_{\max})^{-1}$.

Immune-pathogen recognition depends both on the binding affinity $E_{\alpha,\theta}(r, v)$ and the encounter rate $\gamma_v(t)$ between an immune receptor $r$ and the antigen $v$ at a given time $t$. The encounter rate $\gamma_v(t)$ depends on the abundance of the antigen and the immune receptor, and hence, can vary during an infection within a host. The probability that a receptor $r$ encounters and binds to an antigen $v$ in a short time interval $[t, t + dt]$ can be expressed by, $\rho(r, v, t)\mathrm{d}t = \gamma_v(t)E_{\alpha,\theta}(r, v)\mathrm{d}t$; a similar notion of encounter rate has been previously used in *Mayer et al., 2016*. A memory response in an individual is triggered through the recognition of an antigen by a circulating memory receptor. If no such recognition occurs during the deliberation time $\tau \approx 1.5 - 5$ days, the immune system initiates a naïve response. Therefore, the probability that an antigen is recognized through a novel naive response $P_{\mathrm{recog.}}^{(0)}$ can be expressed as the probability of the antigen not being recognized $1 - P_{\mathrm{recog.}}^{(m)}$ by an available memory receptor $r_m$ over the deliberation period $\tau$,

$$P_{\mathrm{recog.}}^{(0)}(v) = 1 - P_{\mathrm{recog.}}^{(m)}(r_m, v) = e^{-\int_0^\tau \rho(v,t)\mathrm{d}t} = e^{-E_{\alpha,\theta}(r_m, v)\Gamma(v,\tau)} \tag{6}$$

where $\Gamma(v, \tau) = \int_0^\tau \gamma_v(t)dt$ is the expected number of pathogenic encounters over the deliberation time $\tau$ and depends on the accumulated pathogenic load, as pathogens proliferate in the absence of an effective memory prior to a naive response. Here, we have assumed that the affinity of the memory receptor does not change over the response time, which is a simplification since memory receptor can undergo limited affinity maturation (*Shlomchik, 2018*; *McHeyzer-Williams et al., 2018*). To further simplify, we also assume that the accumulated pathogenic load is independent of the type of the pathogen $\Gamma(v, \tau) \equiv \Gamma(\tau)$. As pathogens evolve away from the primary infector, the binding affinity $E_{\alpha,\theta}(r_m, v)$ of the stored memory receptor $r_m$, and hence, the probability to mount a memory response $P_{\mathrm{recog.}}^{(m)}(r_m, v, \tau)$ decays.

The deliberation time prior to a novel response provides a window for memory to react with an antigen and mount an immune response by initiating an irreversible cascade of downstream events. Although initiation of this pathogenic recognition can be modeled as an equilibrium process, the resulting immune response is a non-equilibrium and an irreversible process, the details of which are not included in our model.

## Decision-making to mount a memory or naive response

In the theory of decision-making, a rational decision-maker chooses between two possible actions $a \in \{\text{naive}, \text{memory}\}$ each contributing a utility $U_a$. If the decision-maker has prior preference for each action, which we denote by the prior probability distribution $Q_0(a)$, its decisions could be swayed by this knowledge. As a result, the constrained decision-maker should choose actions according to an optimized probability density $Q(a)$, which maximizes the expected utility while satisfying constraints due to the prior assumption (*von Neumann and Morgenstern, 1944*; *Ortega and Braun, 2013*),

$$Q(a) = \underset{Q(a)}{\text{argmax}} \left( \sum_a U_a Q(a) - \frac{1}{\beta} D_{KL}(Q(a) || Q_0(a)) \right) \tag{7}$$

Here, $D_{KL}(Q(a) || Q_0(a)) = \sum_a Q(a) \log(Q(a)/Q_0(a))$ is the Kullback-Leibler distance between the rational distribution $Q(a)$ and the prior distribution $Q_0(a)$ and $1/\beta$ is a Lagrange multiplier that constrains the efficacy of a decision-maker to process new information and deviate from its prior assumption. The optimal solution for a rational yet constrained decision follows,

$$Q(a) = \frac{1}{Z} Q_0(a) e^{\beta U_a} \tag{8}$$

where $Z = \sum_a Q_0(a) e^{\beta U_a}$ is a normalization factor. If information processing is highly efficient (i.e. the bias factor $1/\beta \to 0$) the rational decision-maker deterministically chooses the action with the highest utility. On the other hand, if the prior is strong (i.e. $1/\beta \to \infty$), the decision-maker hardly changes its opinion and acts according to its prior belief (i.e. $Q(a) = Q_0(a)$). Moreover, if the prior distribution is uniform across actions (i.e. no prior preference), rational decision maximizes the entropy of the system (*Jaynes, 1957*), resulting in the probability of actions $Q(a) \sim \exp[\beta U_a]$. In our analysis, we consider the case of unbiased maximum entropy solution for decision-making. As a result the probability to utilize memory $Q_{\text{mem.}}$ or naive $Q_{\text{naive}}$ follows,

$$Q_{\text{mem.}} = 1 - Q_{\text{naive}} = \frac{e^{\beta U_{\text{mem}}}}{e^{\beta U_{\text{mem}}} + e^{\beta U_{\text{naive}}}} \tag{9}$$

which is a sigmoidal function, dependent on the utility of each action.

A decision to mount a memory or naive response $Q(a)$ based on their respective utilities (*Equation 8*) should be consistent with the biophysical description of the immune response through recognition of an antigen by either of these cell types (*Equation 6*). By equating these two descriptions of an immune response (*Equation 6*, *Equation 8*), we can specify the utility gain associated with mounting a memory or a naïve response in terms of the biophysics and kinetics of receptor-antigen interactions,

$$
\begin{aligned}
Q_{\text{mem.}} = P_{\text{recog.}}^{(m)}(r_m, \upsilon) \quad &\longrightarrow \quad \frac{e^{\beta U_{\text{mem}}}}{e^{\beta U_{\text{mem}}} + e^{\beta U_{\text{naive}}}} = 1 - e^{-E_{\alpha,\theta}(r_m, \nu)\Gamma(\nu, \tau)} \\
&\longrightarrow \quad \beta(U_{\text{mem.}} - U_{\text{naive}}) = \log \left[ e^{E_{\alpha,\theta}(r_m, \nu)\Gamma(\nu, \tau)} - 1 \right]
\end{aligned}
\tag{10}
$$

Importantly, in the regime that memory is efficient and being utilized to mount a response (i.e. a low chance for naive recognition: $P_{\text{recog.}}^{(0)} = e^{-E(\upsilon)\Gamma(\upsilon, \tau)} \ll 1$), the sigmoid form for decision to use memory (*Equation 9*) is dominated by an exponential factor. Therefore, the utility gain by a memory or a naïve response to an evolved antigen $\upsilon_i$ at an antigenic distance $d_i = \|\upsilon_i - \upsilon_0\|$ from the memory receptor's cognate antigen $\upsilon_{r_m}^* \equiv \upsilon_0$ follows (see Materials and methods),

$$
\begin{aligned}
U_{\text{mem}}(\|\upsilon_i - \upsilon_0\|; \alpha, \theta) &= U_{\text{naive}} + E_{\alpha,\theta}(r_m, \nu_i) \\
&= -\Omega(\Gamma_\tau) + E_{\alpha,\theta}(\|\nu_i - \nu_0\|)
\end{aligned}
\tag{11}
$$

Here, we introduce the cost for deliberation $\Omega(\Gamma_\tau)$ as the negative utility of the naive response $U_{\text{naïve}}$. Deliberation cost $\Omega(\Gamma_\tau)$ is a monotonically increasing function of the cumulative pathogen load $\Gamma_\tau$ and reflects the damage (cost) incurred by pathogens as they proliferate during the deliberation time $\tau$ prior to activation of the novel naive response; see *Figure 1*. It is important to note that the difference in the memory and the naïve utility $\Delta U = U_{\text{mem}} - U_{\text{naive}}$ determines the decision to mount either of these responses.

The same consistency criteria between decision-making (*Equation 8*) and cellular recognition (*Equation 6*) indicates that the information processing factor $\beta$ in *Equation 8* should be equal to the accumulated pathogenic load $\Gamma(\tau)$ during the deliberation period $\tau$: $\beta = \Gamma(\tau)$. A longer deliberation, which on one hand leads to the accumulation of pathogens, would allow the immune system to exploit the utility of a usable memory (i.e. process information), even if the memory has only a slight advantage over a responsive naive receptor. As a result, we refer to $\beta$ as the *deliberation factor*. Moreover, this analogy relates the efficacy of information processing $\beta$, which plays the role of

inverse temperature in thermodynamics, and the total accumulated pathogenic load $\Gamma(v, \tau)$, which acts as the sample size for memory receptors as they encounter and accumulate information about pathogens. Interestingly, previous work has drawn a similar correspondence between the inverse temperature in thermodynamics and the effect of sample size on statistical inference *LaMont and Wiggins, 2019*.

The deliberation factor in the immune system should be bounded $\beta \leq \beta_{\max}$ in order for the organism to survive new infections by mounting a novel response that can suppress an exponentially replicating pathogen before it overwhelms the host. Using this bound, we define rescaled deliberation factor $\hat{\beta} \equiv \beta/\beta_{\max} \leq 1$ to characterize the kinetics of an immune response in a dimensionless fashion.

It should be noted that our decision-making formalism assumes that if memory is available, it can be utilized much more efficiently and robustly than a naive response. Therefore, we do not consider scenarios where memory and naive responses are equally involved in countering an infection—a possibility that could play a role in real immune responses. Nonetheless, since such mixed responses are relatively rare, we expect that including them in our model would only result in a slightly different interpretation of the deliberation factor β and should not qualitatively impact our results.

If the immune system decides to mount a memory response against an evolved antigen $v_i$, the binding profile of memory against the target pathogen remains unchanged and equal to the profile $E_{\alpha,\theta}(r_{v_0}, v)$ against the primary infection $v_0$. However, if the immune system mounts a naïve response, a new memory receptor $r_{v_i}$ would be generated with a binding profile $E_{\alpha,\theta}(r_{v_i}, v)$, centered around the latest infection $v_i$. As a result, the expected binding profile $\overline{E_{\alpha,\theta}^{(i)}}(v)$ at the $i^{th}$ round of infection is an interpolation between the profiles associated with memory and naive response, weighted by the likelihood of each decision (*Equation 6*),

$$\overline{E_{\alpha,\theta}^{(i)}}(v) = P_{\text{recog.}}^{(m)}(r_{v_0}, v_i)E_{\alpha,\theta}(r_{v_0}, v) + P_{\text{recog.}}^{(0)}(v_i)E_{\alpha,\theta}(r_{v_i}, v) \tag{12}$$

The expected binding profile at the $i^{th}$ round of infection $\overline{E_{\alpha,\theta}^{(i)}}(v)$ (*Equation 12*) deviates from the optimal profile centered around the infecting pathogen $E_{\alpha,\theta}(r_{v_i}, v)$ (i.e. memory profile stored following a novel response); see *Figure 1*. This deviation arises because an energetically sub-optimal memory response can still be favorable when time is of an essence and the decision has to be made on the fly with short deliberation. This tradeoff between the kinetics and the energetics of immune response results in a *non-equilibrium decision-making Grau-Moya et al., 2018* by the immune system. In analogy to non-equilibrium thermodynamics, we express this deviation as a dissipative cost of memory response $K_{\text{diss}}(t_i; \alpha, \theta)$ at the $i^{th}$ round of infection (time point $t_i$), which we quantify by the Kullback-Leibler distance between the expected and the optimal binding profiles, in units of the deliberation factor β,

$$\begin{aligned} K_{\text{diss}}(t_i; \alpha, \theta) &= \frac{1}{\beta} D_{KL}\left(\overline{E_{\alpha,\theta}^{(i)}}(v) || E_{\alpha,\theta}(r_{v_i}, v)\right) \\ &= \frac{1}{\beta} \sum_{\text{antigens}:v} \overline{E_{\alpha,\theta}^{(i)}}(v) \log\left[\frac{\overline{E_{\alpha,\theta}^{(i)}}(v)}{E_{\alpha,\theta}(r_{v_i}, v)}\right] \end{aligned} \tag{13}$$

where we ensure that binding profiles are normalized over the space of antigens. The dissipation $K_{\text{diss}}$ measures the sub-optimality (cost) of the mounted response through non-equilibrium decision-making and quantifies deviation from an equilibrium immune response *Grau-Moya et al., 2018*.

An optimal memory strategy should be chosen such that it maximizes the expected utility of the immune response $\langle U \rangle = U_{\text{mem}}P_{\text{recog.}}^{(m)} + U_{\text{naive}}P_{\text{recog.}}^{(0)}$, while minimizing the dissipation cost due to the non-equilibrium response $K_{\text{diss}}$, over the lifetime of an organism. To infer an optimal strategy, we introduce net utility $U_{\text{net}}$ that accounts for the tradeoff between the expected utility and dissipation at a given round of infection at time point $t_i$,

$$U_{\text{net}}(t_i; \alpha, \beta, \theta) = \langle U_{\alpha,\beta,\theta}(t_i) \rangle - K_{\text{diss}}(t_i; \alpha, \theta) \tag{14}$$

Net utility can be interpreted as the extracted (information theoretical) work of a rational decision-maker that acts in a limited time, and hence, is constantly kept out of equilibrium (*Grau-Moya et al., 2018*). We infer the optimal memory protocol (i.e. the optimal memory specificity $\alpha^*$

and deliberation factor $\beta^*$) by maximizing the total net utility of memory responses throughout the lifetime of an organism,

$$(\alpha^*, \beta^*) = \underset{\alpha, \beta}{\mathrm{argmax}} \sum_{i:\text{infections}} U_{\text{net}}(t_i; \alpha, \beta, \theta). \tag{15}$$

While we do not model time limits to memory, we effectively model only one memory at a time. This effect is the consequence of modeling the memory as only being beneficial until a novel immune response is triggered resulting in the storage of an updated memory centered around a more recent antigen (*Figure 1*). After such an update, the old memory is no longer relevant as antigens have drifted away.

In our model, the characteristic time for a novel response (and memory update) is set by the expected antigenic divergence (*Figure 2*). Accordingly, cross-reactivity of memory is optimized so that the organism can mount effective responses against evolved forms of antigens in this window of time. However, if the lifetime of memory were to be shorter than this characteristic time of memory update, we expect the organism to store more specific memory since this memory would be utilized to counter a more limited antigenic evolution before it is lost. In other words, the shorter of either the memory lifetime or the characteristic time for memory updates determines the optimal cross-reactivity for immune memory.

## Acknowledgements

We are thankful to Sarah Cobey, Sid Goyal, Lauren McGough, and Josh Plotkin for insightful comments on the manuscript. This work has been supported by the DFG grant (SFB1310) for Predictability in Evolution and the MPRG funding through the Max Planck Society. OHS also acknowledges funding from Georg-August University School of Science (GAUSS) and the Fulbright foundation.

## Additional information

### Competing interests

Armita Nourmohammad: Reviewing editor, *eLife*. The other author declares that no competing interests exist.

### Funding

| Funder | Grant reference number | Author |
| --- | --- | --- |
| Deutsche Forschungsgemeinschaft | SFB1310 | Armita Nourmohammad |
| Max Planck Society | MPRG funding | Armita Nourmohammad |
| University of Washington | Royalty Research Fund: A153352 | Armita Nourmohammad |

The funders had no role in study design, data collection and interpretation, or the decision to submit the work for publication.

### Author contributions

Oskar H Schnaack, Conceptualization, Resources, Formal analysis, Validation, Investigation, Methodology, Writing - original draft, Project administration, Writing - review and editing; Armita Nourmohammad, Conceptualization, Resources, Formal analysis, Supervision, Funding acquisition, Validation, Investigation, Visualization, Methodology, Writing - original draft, Project administration, Writing - review and editing

### Author ORCIDs

Armita Nourmohammad (iD) https://orcid.org/0000-0002-6245-3553

#### Decision letter and Author response
Decision letter https://doi.org/10.7554/eLife.61346.sa1
Author response https://doi.org/10.7554/eLife.61346.sa2

## Additional files

### Supplementary files
• Transparent reporting form

### Data availability
All codes for simulations and numerical analysis can be found at: https://github.com/StatPhysBio/ImmuneMemoryDM (copy archived at https://archive.softwareheritage.org/swh:1:rev:c71f7ab35ebcdd251e4a26fdf9628386fe404e86).

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
