## [Decision Letter]

**Acceptance summary:**

This paper lies out an interesting framework to think about the consequences of immune memory. The central tension is one between memory cells with high affinity for a narrow range of antigens and lower affinity but broader spectrum that can deal with evolving pathogens better. The reviewers appreciated the combination of abstraction, involving thermodynamic and information theoretic frameworks and the grappling with specifics of immune memory.

**Decision letter after peer review:**

Thank you for submitting your article "Optimal evolutionary decision-making to store immune memory" for consideration by *eLife*. Your article has been reviewed by 2 peer reviewers, and the evaluation has been overseen by a Reviewing Editor and Naama Barkai as the Senior Editor. The following individual involved in review of your submission has agreed to reveal their identity: Kayla Sprenger (Reviewer #2).

Essential revisions:

1. As promised in the introduction, the authors could comment on how their approach could be applied to memory T cells in future work, even if the dynamics of such memory is different. (Reviewer 1)

2. Improve Figure 1 so it defines the model more comprehensively without the reader having to consult the Methods section. (Reviewer 1)

3. Cross-reactive Abs are observed to take long to evolve. But the authors results seem to suggest that much cross-reactivity is evolved early in the affinity maturation. The authors should clarify. (Reviewer 2)

4. How would the assumed lifetime of memory B cells affect your results? It appears you have assumed that memory B cells persist through the lifespan of the organism. A discussion of how your results would change if memory lasted for less time would be useful. (Reviewer 2)

*Reviewer #1:*

In this article, Schnaack and Nourmohammad explore the dynamic constraints in triggering optimal immune responses to eradicate infections. The optimization balances the need for higher affinity (higher specificity/efficacy in clearing pathogen) and cross-reactivity (better response to evolving pathogens). The key paradox being addressed here is that, depending on the speed of evolution of pathogens, the immune system can tune the specificity of the lymphocytes' receptor that are selected e.g. by adjusting the size of low-affinity of the memory lymphocyte compartment.

Given that the formalism is quite abstract (e.g. definition of antigenic distance in the shape space), it is hard to assess how experimentally testable the results are. The authors make a good attempt at discussing their main insight and it is certainly thought-provoking: they found that the number of exposures to a pathogen, as it relates to the age span of the organism under consideration, is a critical parameter to decide the amount of cross-reactivity stored in memory leukocytes. This is the strongest insight as it relates to experimental results.

There are theoretical surprises as well: the bimodality of receptor specificities that get selected when pathogens are very diverse at the antigen level is thought-provoking. The authors do point out that this may relate to experimental observations for B cells (mixed populations are selected with or without class switching).

Overall, this is a very well written and insightful manuscript leveraging results from non-equilibrium statistical physics and accounting for varied strategies for the immune system.

*Reviewer #2:*

This manuscript features a novel mathematical model for investigating how the immune system optimally stores memory of B cell receptor-pathogen interactions in order to maximize protection against future pathogens with diverse evolutionary rates. Results support recent experimental findings that B cell differentiation into memory cells is strongly regulated during the affinity maturation process and that the kinetics and energetics of an immune response are simultaneously optimized to ensure an effective response. Unique insights are also provided into the immunological phenomenon of original antigenic sin, such as the effect on this phenomenon of organismal lifetime, which is also explored in the context of optimal memory storage strategies.

Strengths:

The presented mathematical framework is rigorously constructed such that meaningful insights can be gleaned into the workings of the adaptive immune response to evolving pathogens. The framework combines fundamental concepts from information processing and equilibrium and non-equilibrium thermodynamics with concepts from probability and statistics in a unique and thoughtful way. The conclusions appear to be well-aligned with recent experimental findings, providing validity for the model and for the subsequent predictions that are made on optimal memory storage strategies for organisms with varying lifespans.

The results provide useful insights into longstanding questions in immunology, such as whether cell fate decisions on memory B cell differentiation are regulated during the process of affinity maturation, and into the origins of original antigenic sin from an immune response perspective and potential mitigation strategies. With regards to the former point, the model accurately reproduces recent experimental findings showing differentiation into memory B cells during affinity maturation is indeed highly regulated. This thus sets a bar for future computational models of immunological memory processes and affinity maturation to incorporate this feature, rather than assuming differentiation into memory B cells is stochastic and carried out at a constant rate throughout affinity maturation, which is currently a common assumption.

Broad parameter regimes are explored, rendering the findings potentially relevant for infection scenarios with diverse pathogens.

Weaknesses:

Typically, cross-reactivity or equivalently breadth takes a long time to evolve, as evidenced by the fact that broadly neutralizing antibodies (bnAbs) arise only after many years of infection (or re-infection) by an evolving pathogen. Arguments are made by the authors that memory B cells are preferentially produced early on in the affinity maturation process, and that memory B cells are also preferentially stored with intermediate cross-reactivity, which would seem to imply that a good deal of cross-reactivity can be evolved early on in the maturation process. These arguments would seem to be at odds with the concept of bnAb evolution and thus warrant some clarity.

Two additional points that may warrant some clarity are:

(1) How much the results, especially in the context of organisms of varying life spans, depend on the presumed assumptions that memory B cells have a lifespan that persists throughout the lifetime of the host (seems to still be somewhat of an open question) and that immunological memory does not decay with time, and;

(2) the role of B cell precursor frequencies in the decision-making process of mounting a memory versus naïve B cell response. The authors define an effective deliberation time for the naïve B cell population to reach a level of activity that is similar to the memory B cell population, based on the argument that memory B cells "can respond quicker and in larger numbers" than naïve B cells. However, one could imagine a scenario where the memory B cell response is quickly outcompeted and overshadowed by the naïve B cell response due to the high precursor frequencies of the naïve B cells.

The impact of the paper could potentially be heightened if some discussion of how the principles gleaned on optimal immune memory strategies could be translated to, e.g., vaccine design against fast-evolving pathogens.

General comments:

1. Lines 60-61, It is stated that "as in most molecular interactions, immune pathogen recognization is cross-reactive". I am confused by this statement, as many molecular interactions are indeed not cross-reactive (e.g., lock-and-key binding of enzymes and ligands, etc.). Immune-pathogen recognition would also not typically appear to be cross-reactive unless the pathogen is highly mutable or there have been multiple infections of an evolved pathogen, so this sentence further confuses me. Please note that if the sentence is kept as is, I believe the authors meant to use "recognition" instead of "recognization".

2. Besides the concept of original antigenic sin, could the concept of immune imprinting where immune memory is biased over the lifetime of an organism also be captured by or incorporated into this model somehow?

3. As it appears to be defined, antigenic divergence characterizes two distinct infections by a given pathogen. How relevant is this model and its results for a pathogen like HIV that mutates within its host, where a range of antigenic distances/divergence values quickly become relevant for a single infection?

4. Line 142: in what kinds of scenarios (or against what types of pathogens), might prior preferences be important to consider? To clarify, all of the analyses carried out here assume no prior preferences?

5. In regard to Equation 2, it would seem that the same maximum net utility value could be obtained with either a particularly high expected utility or a particularly low Kdiss. Would the optimal memory protocol look different in these two cases, despite them having the same net utility value? Perhaps this is already addressed, but the answer is not immediately clear to me.

6. Lines 215-218, it is stated that "This optimization [for the case of moderately evolving pathogens] results in a smaller deliberation factor 𝛽 compared to the scenario with slowly evolving pathogens, yet a long enough deliberation to allow the energetically suboptimal memory to react to an infection". This appears to be true for a range of δ about 0.1-0.5. Yet, still within in the defined range of δ for moderately evolving pathogens, δ values between about 0.6 and 0.8 for the lower amplitude cases appear to result in 𝛽 values above those for slowly evolving pathogens. Can the authors please clarify this?

7. Line 104, it is not clear what is meant by the statement that "Physico-chemical constraints in protein structures can introduce a tradeoff between immune receptors' affinity and cross-reactivity". Is this tradeoff not determined by the antigenic divergence that an immune system encounters upon a new infection compared to a past infection?

8. Lines 251-254, the last sentence seems to imply that a moderate amount of cross-reactivity or equivalently breadth is achieved early on in the affinity maturation process. This conflicts with my understanding of the evolution of broadly neutralizing antibodies, which typically arise only after many years of infection/pathogen mutation within a host. Can the authors please comment on this? In addition, I was under the impression that broadly neutralizing antibodies are typically class-switched, i.e., not of the IgM type. Is their evidence that the cross-reactivity of IgM receptors produced early on in affinity maturation is really effective at 'countering evolving pathogens' (lines 279-282)?

9. In the last Results section on the effect of infection frequency, and perhaps in general throughout the manuscript, is the assumption made that memory B cells persist for the entire lifetime of an organism? Some studies have placed the half-life of memory B cells to be only between 8 and 10 weeks, and others up to or possibly beyond 2 years, and still others for the lifetime of the host but requiring constant renewal through antigen-specific stimulation. How might changes in the expected lifetime of memory B cells affect the optimal memory strategies that are presented?

---

## [Author Response]

Essential revisions:1. As promised in the introduction, the authors could comment on how their approach could be applied to memory T cells in future work, even if the dynamics of such memory is different. (Reviewer 1)

We have now included a paragraph in the Discussion section highlighting the parallels between T cell and B cell memory generation (lines 436-452). Of course, as the reviewers have pointed out the mechanisms are distinct but conceptually one may be able to give a similar evolutionary rationale for understanding memory differentiation in these B-cells and T-cells.

From the text:

“Although mechanistically distinct from B-cells, T-cells also differentiate into effector and memory in response to infections. […] Taken together, there are parallels between differentiation of T-cells and B-cells to memory, and it will be interesting to investigate the advantages of storing cross-reactive (and plausibly low-affinity) T-cell memory as a strategy to counter evolving pathogens.”

2. Improve Figure 1 so it defines the model more comprehensively without the reader having to consult the Methods section. (Reviewer 1)

We have extended Figure 1 to indicate more details about the model in the schematic.

3. Cross-reactive Abs are observed to take long to evolve. But the authors results seem to suggest that much cross-reactivity is evolved early in the affinity maturation. The authors should clarify. (Reviewer 2)

It is true that BnAbs, which are known for their cross-reactivity, take long to evolve within individuals. However, it should be noted that BnAb’s breadth is often achieved by targeting highly vulnerable regions of the virus, including the CD4 binding site in HIV or the influenza stem. In a sense, BnAbs achieve their breadth by specifically targeting conserved regions of a virus that are shared among a broad panel of strains. This is of course not true for all BnAbs and some are broad in a true sense. Still, we argue that BnAbs are more of an exception than a rule when thinking about cross-reactive interactions. As we are now clarifying in the text, we consider cross-reactivity as a feature that makes naive (or pre-class switched) antibodies flexible to interact with different pathogenic targets. With maturation, a receptor’s affinity increases and it becomes more rigid and specific to its cognate antigen (Fern´andez-Quintero et al. (2020); Li et al. (2003); Mishra and Mariuzza (2018); Wedemayer et al. (1997a)).

We are now clarifying these points in the main text (Lines 111-117) and in the Methods (lines 545560) when discussing our model of receptors’ affinity and cross-reactivity.

4. How would the assumed lifetime of memory B cells affect your results? It appears you have assumed that memory B cells persist through the lifespan of the organism. A discussion of how your results would change if memory lasted for less time would be useful. (Reviewer 2)

As the reviewer has pointed out, memory B-cells can persist for a very long time (e.g. over 50 years after smallpox vaccination). However, long-term memory does not necessarily imply that individual cells have a long life span, but rather it can be due to the persistence of (renewing) clones specific to a given antigen. Indeed, the mechanisms to maintain a long-lived immune memory are not well understood. Notably, experiments in mice have shown that a memory B-cell’s half-life can exceed the lifespan of the organism (Jones et al. (2015)) but we still lack such estimates for memory B-cells in humans.

While we do not model time limits to memory, we effectively model only one dominant memory at a time. This process is the consequence of modeling the memory as only being beneficial until a novel immune response is triggered resulting in the storage of an updated memory centered around a more recent antigen. After such update, the old memory is no longer relevant as antigens have drifted away.

In our current setup, the characteristic time for a novel response (and memory update) is set by the expected antigenic divergence. Accordingly, cross-reactivity of memory is optimized so that the organism can mount effective responses against evolved forms of antigens in this window of time. However, if the lifetime of memory were to be shorter than this characteristic time of memory update, we expect the organism to store more specific memory since this memory would be utilized to counter a more limited antigenic evolution before it is lost. In other words, the shorter of either the memory lifetime or the characteristic time for memory updates determines the optimal cross-reactivity for immune memory.

We are now discussing these effects in the manuscript (lines 717-729).

Reviewer #2:1. The role of B cell precursor frequencies in the decision-making process of mounting a memory versus naïve B cell response. The authors define an effective deliberation time for the naïve B cell population to reach a level of activity that is similar to the memory B cell population, based on the argument that memory B cells "can respond quicker and in larger numbers" than naïve B cells. However, one could imagine a scenario where the memory B cell response is quickly outcompeted and overshadowed by the naïve B cell response due to the high precursor frequencies of the naïve B cells.

We thank the reviewer for raising this interesting question. A series of experiments have investigated different aspects of this question (Blanchard-Rohner et al. (2009); Tangye and Hodgkin (2004); Tangye et al. (2003)). In brief:

Previous experiments indicate that memory cells are fast responders and initiate cell division about *τ*_0_ ≈ 1−2 days before na¨ıve cells (Blanchard-Rohner et al. (2009); Tangye and Hodgkin (2004); Tangye et al. (2003)). Moreover, the number of memory cells that are recruited to proliferate and differentiate to effector cells is *b* ≈ 2−3 times larger than the number of na¨ıve cells (Tangye and Hodgkin (2004); Tangye et al. (2003)). Once recruited, however, memory and na¨ıve cells have approximately a similar doubling time of about *t_1/2_* ≈ 0.5 − 2 days (Macallan et al. (2005); Tangye et al. (2003)). Putting these together, we can define an effective deliberation time *τ* for the na¨ıve population to reach an activity level (i.e., a population size) comparable to the memory. Assuming an exponential growth during the early stages of memory and na¨ıve proliferation, the deliberation time can be estimated in terms of the kinetic factors by *τ* = *τ*_0_ + *t_1/2_* ln*b/*ln2 and it is within a range of *τ* ≈ 1.5 − 5 days. Therefore, given the experimental evidence, if memory is available and is activated, it is unlikely that it would be outcompeted and overshadowed by the naive B cell response.

These arguments were previously only presented in the Methods section (lines 521-537). We have now included these arguments in the main text (Lines 137-142) to better justify our assumptions.

2. The impact of the paper could potentially be heightened if some discussion of how the principles gleaned on optimal immune memory strategies could be translated to, e.g., vaccine design against fast-evolving pathogens.

Thanks for bringing up this question. We have added a discussion on this matter to the manuscript (lines 388-406).

General comments:3. Lines 60-61, It is stated that "as in most molecular interactions, immune pathogen recognization is cross-reactive". I am confused by this statement, as many molecular interactions are indeed not cross-reactive (e.g., lock-and-key binding of enzymes and ligands, etc.). Immune-pathogen recognition would also not typically appear to be cross-reactive unless the pathogen is highly mutable or there have been multiple infections of an evolved pathogen, so this sentence further confuses me. Please note that if the sentence is kept as is, I believe the authors meant to use "recognition" instead of "recognization".

We have removed the misleading part of the sentence.

4. As it appears to be defined, antigenic divergence characterizes two distinct infections by a given pathogen. How relevant is this model and its results for a pathogen like HIV that mutates within its host, where a range of antigenic distances/divergence values quickly become relevant for a single infection?

We expect the model to still hold for the intra-host evolution of pathogens like HIV. Indeed, Immune response to chronic HIV involves turnover of B-cell lineages over time (Nourmohammad et al. (2019)), which one may argue that resembles elicitation of novel responses to diverged antigens, against which prior memory is ineffective. It would be interesting to see how the constant pressure from HIV on the host’s immune system and the gradual depletion of helper T-cells would impact the efficacy of memory over short, intermediate, and long time scales.

We have included these discussions to the manuscript (Lines 426-435).

5. Line 142: in what kinds of scenarios (or against what types of pathogens), might prior preferences be important to consider? To clarify, all of the analyses carried out here assume no prior preferences?

The reviewer is correct that we use no prior preferences in our analyses, and we are now explicitly stating this fact in the manuscript (Lines 156-158).

The implications of this question are very interesting. We expect (in principle) that extreme choices of priors impact the outcome of our model. One can however turn this question around and ask whether such priors can emerge as optimal strategies in light of immune-pathogen coevolution. One possibility to imagine is whether there is an evolutionary advantage to optimize immune machinery (over evolutionary time scales) so that it generates repertoires that can respond more efficiently to commonly observed pathogens. Among other factors, the outcome of this optimization depends on the space of pathogens, the diversity of receptor repertoires, and the constraints imposed by self/ non-self discrimination (see e.g. Mayer et al. (2015, 2019)). It would be very interesting to study how such priors could naturally emerge in the context of immune-pathogen coevolution, but this topic is beyond the scope of this manuscript.

6. In regard to Equation 2, it would seem that the same maximum net utility value could be obtained with either a particularly high expected utility or a particularly low Kdiss. Would the optimal memory protocol look different in these two cases, despite them having the same net utility value? Perhaps this is already addressed, but the answer is not immediately clear to me.

The reviewer is correct that dissipation and the expected utility could in principle compensate each other and result in degenerate states with the same optimum net utility. If such degenerate solutions were to be available, we expect to see multiple optima for the net utility function in the specificity / deliberation parameter space (*α*^∗^*,β*^∗^). However, as shown in Figure 2, for each value of antigenic divergence, we only obtain a single global optimum and do not see any degeneracy in the solutions.

7. Lines 215-218, it is stated that "This optimization [for the case of moderately evolving pathogens] results in a smaller deliberation factor β compared to the scenario with slowly evolving pathogens, yet a long enough deliberation to allow the energetically suboptimal memory to react to an infection". This appears to be true for a range of δ about 0.1-0.5. Yet, still within in the defined range of δ for moderately evolving pathogens, δ values between about 0.6 and 0.8 for the lower amplitude cases appear to result in β values above those for slowly evolving pathogens. Can the authors please clarify this?

We have adjusted the regime to 20% − 60% to avoid any confusion.

8. Line 104, it is not clear what is meant by the statement that "Physico-chemical constraints in protein structures can introduce a tradeoff between immune receptors' affinity and cross-reactivity". Is this tradeoff not determined by the antigenic divergence that an immune system encounters upon a new infection compared to a past infection?

This tradeoff is rooted in the biophysics of protein (and specifically antibody) structures. There has been accumulating evidence that as B-cell receptors maturate, hypermutations strengthen the binding affinity of the corresponding antibodies and at the same time make their structures more rigid and specific to their cognate antigens (Fern´andez-Quintero et al. (2020); Frank (2002); Li et al. (2003); Mishra and Mariuzza (2018); Wedemayer et al. (1997b); Wu et al. (2017)). In contrast, naive (or pre-class switched) antibodies are more flexible and they can interact with different pathogenic targets (i.e., they are more cross-reactive). Taken together, these structural studies suggest that the physico-chemical constraints in protein structures of antibodies cause a tradeoff between affinity and cross-reactivity, at least for a broad class of such molecules. Of course, there are always exceptions to this rule and further studies are necessary to better understand the biophysical constraints on antibody structures.

These points were previously discussed in the Methods (lines 545-560) and we have now included these statement earlier in the text when the model is introduced (Lines 108-111).

References:

G. Blanchard-Rohner, A. S. Pulickal, C. M. Jol-van der Zijde, M. D. Snape, and A. J. Pollard. Appearance of peripheral blood plasma cells and memory B cells in a primary and secondary immune response in humans. Blood, 114(24):4998–5002, Dec. 2009.

M. L. Fern´andez-Quintero, J. R. Loeffler, L. M. Bacher, F. Waibl, C. A. Seidler, and K. R. Liedl. Local and Global Rigidification Upon Antibody Affinity Maturation. Front. Mol. Biosci., 7:182, Aug. 2020. ISSN 2296-889X.

S. A. Frank. Immunology and Evolution of Infectious Disease. Princeton University Press, Princeton (NJ), 2002.

J. A. Greenbaum, M. F. Kotturi, Y. Kim, C. Oseroff, K. Vaughan, N. Salimi, R. Vita, J. Ponomarenko, R. H. Scheuermann, A. Sette, and B. Peters. Pre-existing immunity against swine-origin H1N1 influenza viruses in the general human population. Proceedings of the National Academy of Sciences, 106(48):20365–20370, Dec. 2009. ISSN 0027-8424, 1091-6490.

D. D. Jones, J. R. Wilmore, and D. Allman. Cellular Dynamics of Memory B Cell Populations: IgM ^+^ and IgG ^+^ Memory B Cells Persist Indefinitely as Quiescent Cells. J.I., 195(10):4753–4759, Nov. 2015. ISSN 0022-1767, 1550-6606. doi: 10.4049/jimmunol.1501365. URL http://www.jimmunol.org/lookup/doi/10.4049/jimmunol. 1501365.

C. Kim and M. A. Williams. Nature and nurture: T-cell receptor-dependent and T-cell receptor-independent differentiation cues in the selection of the memory T-cell pool: TCR-driven hierarchical differentiation of CD4^+^ T cells. Immunology, 131(3):310–317, Nov. 2010. ISSN 00192805.

Y. Li, H. Li, F. Yang, S. J. Smith-Gill, and R. A. Mariuzza. X-ray snapshots of the maturation of an antibody response to a protein antigen. Nat Struct Mol Biol, 10(6):482–488, June 2003. ISSN 1545-9993, 1545-9985.

D. C. Macallan, D. L. Wallace, Y. Zhang, H. Ghattas, B. Asquith, C. de Lara, A. Worth, G. Panayiotakopoulos, G. E. Griffin, D. F. Tough, and P. C. L. Beverley. B-cell kinetics in humans: rapid turnover of peripheral blood memory cells. Blood, 105(9):3633–3640, May 2005.

K. Man, M. Miasari, W. Shi, A. Xin, D. C. Henstridge, S. Preston, M. Pellegrini, G. T. Belz, G. K. Smyth, M. A. Febbraio, S. L. Nutt, and A. Kallies. The transcription factor IRF4 is essential for TCR affinity–mediated metabolic programming and clonal expansion of T cells. Nat Immunol, 14(11):1155–1165, Nov. 2013. ISSN 1529-2908, 15292916.

A. Mayer, V. Balasubramanian, T. Mora, and A. M. Walczak. How a well-adapted immune system is organized. Proc. Natl. Acad. Sci. U.S.A., 112(19):5950–5955, May 2015.

A. Mayer, V. Balasubramanian, A. M. Walczak, and T. Mora. How a well-adapting immune system remembers. Proc. Natl. Acad. Sci. U.S.A., 116(18):8815–8823, Apr. 2019.

A. K. Mishra and R. A. Mariuzza. Insights into the Structural Basis of Antibody Affinity Maturation from NextGeneration Sequencing. Front. Immunol., 9:117, Feb. 2018. ISSN 1664-3224.

A. Nourmohammad, J. Otwinowski, M. L uksza, T. Mora, and A. M. Walczak. Fierce Selection and Interference in B-Cell Repertoire Response to Chronic HIV-1. Molecular Biology and Evolution, 36(10):2184–2194, Oct. 2019. ISSN 0737-4038, 1537-1719.

R. Roychoudhuri, D. Clever, P. Li, Y. Wakabayashi, K. M. Quinn, C. A. Klebanoff, Y. Ji, M. Sukumar, R. L. Eil, Z. Yu, R. Spolski, D. C. Palmer, J. H. Pan, S. J. Patel, D. C. Macallan, G. Fabozzi, H.-Y. Shih, Y. Kanno, A. Muto, J. Zhu, L. Gattinoni, J. J. O’Shea, K. Okkenhaug, K. Igarashi, W. J. Leonard, and N. P. Restifo. BACH2 regulates CD8^+^ T cell differentiation by controlling access of AP-1 factors to enhancers. nature immunology, 17 (7):12, July 2016.

R. L. Rutishauser, G. A. Martins, S. Kalachikov, A. Chandele, I. A. Parish, E. Meffre, J. Jacob, K. Calame, and S. M. Kaech. Transcriptional Repressor Blimp-1 Promotes CD8^+^ T Cell Terminal Differentiation and Represses the Acquisition of Central Memory T Cell Properties. Immunity, 31(2):296–308, Aug. 2009. ISSN 10747613.

L. K. Selin, M. Cornberg, M. A. Brehm, S.-K. Kim, C. Calcagno, D. Ghersi, R. Puzone, F. Celada, and R. M. Welsh. CD8 memory T cells: cross-reactivity and heterologous immunity. Seminars in Immunology, 16(5):335–347, Oct. 2004. ISSN 10445323.

A. Sette and S. Crotty. Pre-existing immunity to SARS-CoV-2: the knowns and unknowns. Nat Rev Immunol, 20

(8):457–458, Aug. 2020. ISSN 1474-1733, 1474-1741.

S. G. Tangye and P. D. Hodgkin. Divide and conquer: the importance of cell division in regulating B-cell responses. Immunology, 112(4):509–520, Aug. 2004.

S. G. Tangye, D. T. Avery, E. K. Deenick, and P. D. Hodgkin. Intrinsic differences in the proliferation of naive and memory human B cells as a mechanism for enhanced secondary immune responses. J. Immunol., 170(2):686–694, Jan. 2003.

G. J. Wedemayer, P. A. Patten, L. H. Wang, P. G. Schultz, and R. C. Stevens. Structural Insights into the Evolution of an Antibody Combining Site. Science, 276(5319):1665–1669, June 1997a. ISSN 00368075, 10959203. doi: 10.1126/science.276.5319.1665. URL https://www.sciencemag.org/lookup/doi/10.1126/science.276.5319. 1665.

G. J. Wedemayer, P. A. Patten, L. H. Wang, P. G. Schultz, and R. C. Stevens. Structural insights into the evolution of an antibody combining site. Science, 276(5319):1665–1669, June 1997b.

N. C. Wu, G. Grande, H. L. Turner, A. B. Ward, J. Xie, R. A. Lerner, and I. A. Wilson. in vitro evolution of an influenza broadly neutralizing antibody is modulated by hemagglutinin receptor specificity. Nat Commun, 8(1): 15371–12, May 2017.